# Weakened AMOC related to cooling and atmospheric circulation shifts in the last interglacial Eastern Mediterranean

Elan J. Levy [1,2] ✉, Hubert B. Vonhof [1], Miryam Bar-Matthews[2], Alfredo Martínez-García [1], Avner Ayalon[2], Alan Matthews[3], Vered Silverman[4], Shira Raveh-Rubin [4], Tami Zilberman[2], Gal Yasur [2], Mareike Schmitt[1] & Gerald H. Haug[1,5]

There is limited understanding of temperature and atmospheric circulation changes that accompany an Atlantic Meridional Overturning Circulation (AMOC) slowdown beyond the North Atlantic realm. A Peqi'in Cave (Israel) speleothem dated to the last interglacial period (LIG), 129–116 thousand years ago (ka), together with a large modern rainfall monitoring dataset, serve as the base for investigating past AMOC slowdown effects on the Eastern Mediterranean. Here, we reconstruct LIG temperatures and rainfall source using organic proxies ($TEX_{86}$) and fluid inclusion water *d-excess*. The $TEX_{86}$ data show a stepwise cooling from $19.8 \pm 0.2°$ (*ca.* 128–126 ka) to $16.5 \pm 0.6\,°C$ (*ca.* 124–123 ka), while *d-excess* values decrease abruptly (*ca.* 126 ka). The *d-excess* shift suggests that rainfall was derived from more zonal Mediterranean air flow during the weakened AMOC interval. Decreasing rainfall *d-excess* trends over the last 25 years raise the question whether similar atmospheric circulation changes are also occurring today.

Though no absolute consensus has been achieved amongst all data[1,2], mounting evidence suggests that the AMOC has been weakening in recent years[3–5] and/or will weaken in the future[1]. Climate model simulations indicate that a weakened AMOC is associated with shifting atmospheric circulation patterns, together with overall northern hemispheric cooling and changes in rainfall regimes[6,7]. Globally as warm and/or warmer than today[8], the LIG provides an ideal period to study climate dynamics in response to weakened AMOC. The early-LIG at *ca.* 127 ka was marked by an event of decreased oxygenation of deep water by Antarctic Bottom Water (AABW) that coincided with increased North Atlantic Deep Water (NADW) formation[9], followed by abrupt episodes of severely reduced NADW formation suggesting a weakened AMOC[10,11]. Alongside these oceanic disruptions, terrestrial climate change and cooling has been reported in a number of proxy records from central and southern Europe[12–19]. Four rapid centennial-

to-millennial scale North Atlantic LIG cold events (labelled C27, C27′, C27a and C27b) were detected from high resolution paleo-climate records[13]. The North Atlantic C27 cold event was suggested to have followed freshwater discharge from ice sheets and/or outburst flooding from the collapse of the Laurentide ice sheet[13]. Climate model freshwater discharge simulations show AMOC slowdowns in the North Atlantic, at 126.6–126.4 ka and 123.8–123.6 ka, and reveal strong SST cooling, a deepening of the low-pressure system at mid-to-high latitudes, reduction in rainfall, and strengthened westerlies in the North Atlantic at 40°N-50°N[13]. However, while LIG climate models suggest hemispheric-wide climate changes associated with a weakened AMOC, so far, there is little evidence for teleconnection patterns linking AMOC slowdown with Mediterranean cooling.

Israel, situated in the Eastern Mediterranean and Western Asia while influenced by North Atlantic atmospheric conditions,

[1]Department of Climate Geochemistry, Max-Planck Institute for Chemistry, Mainz, Germany. [2]The Geological Survey of Israel, Jerusalem, Israel. [3]The Fredy & Nadine Herrmann Institute of Earth Sciences, The Hebrew University of Jerusalem, Jerusalem, Israel. [4]Department of Earth and Planetary Sciences, Weizmann Institute of Science, Rehovot, Israel. [5]Department of Earth Sciences, ETH Zurich, Zürich, Switzerland. ✉e-mail: elanl@gsi.gov.il

provides an excellent location for investigating climate change resulting from a weakened AMOC[20–24]. It is segmented by Mediterranean, semi-arid, andarid climate regions from north to south. It is also a hot spot for modern climate research as it has been experiencing a reduction in the total rainfall amounts in recent years[25]. Furthermore, the stable H isotope ($\delta^2$H), O isotope ($\delta^{18}$O) and the associated *d-excess* ($\delta^2$H $-8 \cdot \delta^{18}$O) of rainfall is sensitive to moisture source and atmospheric circulation changes in this region[26–29]. Paleo-climate records of rainfall for the LIG exist at Peqi'in and Soreq caves (separated by a distance of ~150 km) in the form of speleothems (stalactites, stalagmites and flowstones)[26] (Fig. 1). These speleothems formed following percolation of rainfall through soil and carbonate host-rock into the caves (as cave drip water), $CO_2$ degassing and subsequent $CaCO_3$ mineralization. The calcite crystals forming the speleothems reveal millennial-scale stable oxygen isotope ($\delta^{18}O_{calcite}$) trends that allowed for the reconstruction of the isotopic evolution of rainfall back in time over the last 250 ka. Fluid inclusions are microscopic-sized pockets of encapsulated cave drip water and provide direct measurements of both $\delta^2$H and $\delta^{18}$O values of accumulated paleo-rainfall[27–29]. Here, we use the $\delta^2$H and $\delta^{18}$O of fluid inclusion water, and corresponding *d-excess* values, from a speleothem at Peqi'in Cave to reconstruct Eastern Mediterranean paleo-rainfall changes[29]. Using a large rainfall monitoring dataset sampled during the last 25 years, together with backward trajectory reconstruction and clustering, we investigate the controlling factors of *d-excess* in modern accumulated rainfall and then use these findings to explain the LIG speleothem derived fluid inclusion *d-excess* record. Additionally, we reconstruct Peqi'in Cave temperatures obtained using the $TEX_{86}$ paleothermometer, which is based on the relative abundance of Glycerol Dialkyl

Glycerol Tetraethers (GDGTs) found in LIG speleothem calcite[30–32]. We use the temperature reconstruction to determine whether the Eastern Mediterranean had undergone cooling, and if yes to what degree, and using the fluid inclusion *d-excess* values we identify associated near surface moisture uptake conditions and shifts in regional rainfall moisture source in the Eastern Mediterranean[29,33,34].

## Results and discussion
### Mid-LIG cooling in the Eastern Mediterranean
The $TEX_{86}$ temperatures at Peqi'in Cave during the early-LIG (*ca.* 128–126 ka) averaged 19.8 ± 0.2 °C (1σ), which is estimated as ~3–4 °C warmer than today (Supplementary Data 1). This warm interval was followed by stepwise cooling to 18.6 ± 0.4 °C at *ca.* 126–125 ka and to 16.5 ± 0.6 °C at *ca.* 124–123 ka. From then until *ca.* 119 ka temperatures increased to >20 °C.

Major LIG temperature changes have also been identified in Europe during the early and mid LIG. The early LIG (early Eemian) was marked by a thermal optimum[35]. For example, pollen derived temperature reconstructions from Sokli (Finland) are 2–3 °C warmer than present during the early-LIG (July temperatures; Fig. 2A)[12,19]. This was followed by a terrestrial cooling event (the Tunturi event), which reduced temperatures by 2.3–3.5 °C, and then a subsequent return to the pre-cooling temperatures. The LIG cooling was associated with the marine C27 event, which likely resulted from an outburst flood via the Hudson Strait[36] and a weakened NADW[10] (Fig. 2B). North Atlantic Sea surface water cooling during this period was also evident in the *Neogloboquadrina pachyderma* (s) coiling ratio[37,38] (Fig. 2B). LIG temperatures derived from fluid inclusion $\delta^2$H at Schrattenkarst Cave's (Switzerland) were up to 4.0 °C warmer compared to the present, but

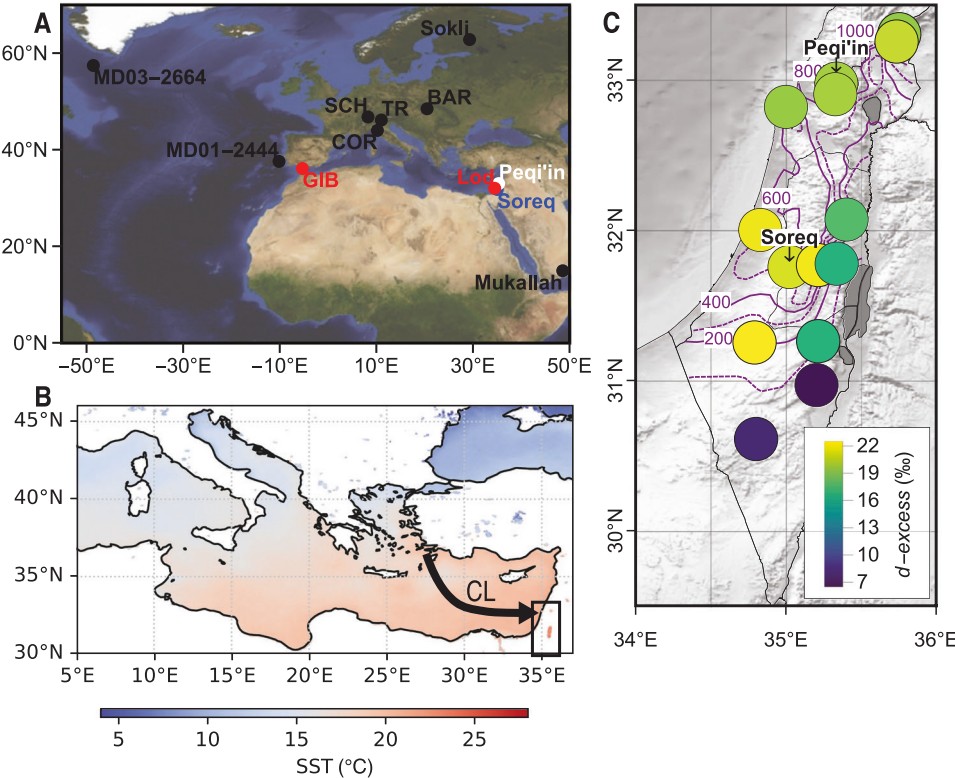

**Fig. 1 | Study region. A** Locations discussed in the study (Black, red, white and blue points). SCH= Schrattenkarst, COR= Corchia, TR= Trentino, BAR= Baradla and GIB= Gibraltar. **B** Median sea surface temperatures (SST; °C) calculated for October to April from 2000 to 2022 (blue to red gradient). For comparison the annual averaged global SST is 16.1 °C (years: 1901–2000). The arrow illustrates a deep Cyprus Low (CL) mid-latitude cyclone trajectory after ref. 22. **C** Relief map with mean annual rainfall isohyets after ref. 34,46. (mm/year; alternating purple regular lines and dashed lines at 100 mm increment's). Locations of Soreq and Peqi'in, and rainfall sampling sites. Disks coloured according to the multiannual accumulated rainfall *d-excess* after ref. 34,46.

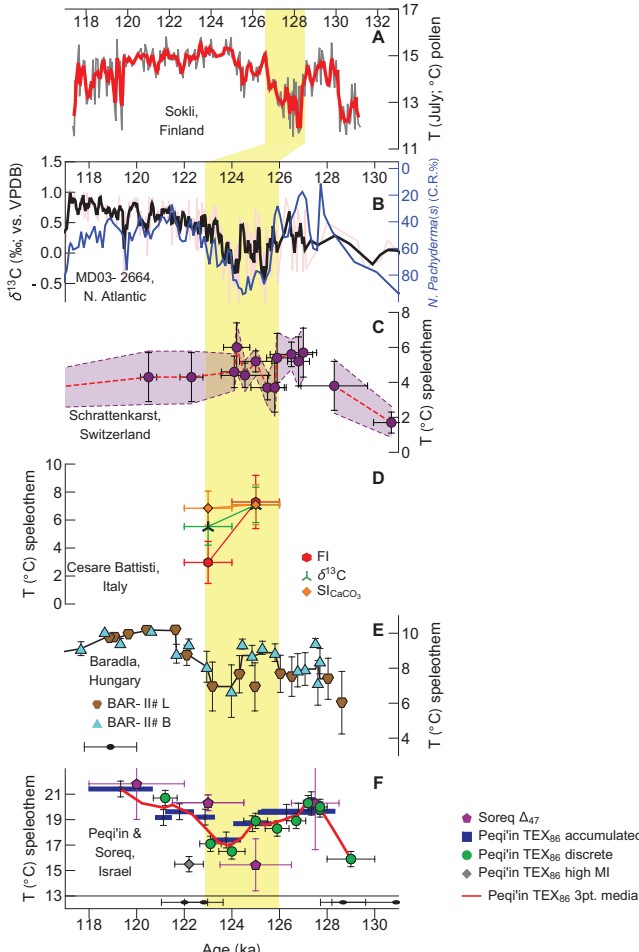

**Fig. 2 | Last interglacial temperature and marine records. A** Pollen reconstructed July temperatures (T) (Sokli, Finland) after ref. 12,19. **B** $\delta^{13}$C(*Cibicidoides weullerstorfi*) (red line) with 3 pt. moving median curve (black line)[10] and *Neogloboquadrina Pachyderma* (s) coiling ratio (blue line) after ref. 37,38. (MD03-2664, North Atlantic). **C** Speleothem fluid inclusion (FI) derived temperatures (Schrattenkarst, Switzerland) after ref. 14. Calculated using various $\delta^2$H/°C calibrations: 0.65‰/°C (purple disks and red dashed line), 0.6‰/°C and 0.7‰/°C (purple dashed lines). Temperature error bars reflect isotope measurement errors, $\delta^{18}$O/annual air temperature slope error, and 1σ of repeated measurements. **D** Temperatures calculated using FI (red hexagons), speleothem $\delta^{13}$C (green triple-cross's) and estimated $CaCO_3$ saturation index (SI; orange diamonds) (Cesare Battisti Cave, Italy) after ref. 17. Temperature error bars consider measurement error, repeated measurement error, and model/calibration equation error. **E** Fluid inclusion $\delta^2$H derived T (cyan triangles and brown polygons for respective BAR-II#L and BAR-II#B speleothems) with 3 pt. moving median (black line) (Baradla, Hungary) after ref. 15,16. Temperature errors calculated from error propagation of analytical and $\delta^2$H/T gradient calibration uncertainties. Sample age uncertainty reflected by black error bar in the lower left. **F** Peqi'in $TEX_{86}$ temperatures from discrete samples (green disks), accumulated samples (blue rectangles), a high methane index sample (grey diamond), and 2 pt. median running curve (red line). Temperature errors were estimated from repeated measurements of a speleothem standard and, for accumulated samples, error propagation. Also shown are clumped isotope ($\Delta_{47}$) temperatures from Soreq Cave speleothems (purple pentagons)[29]. The U-Th ages (± 2σ) for the Peqi'in PEK-9 speleothem are shown in the panel below (black disks). Yellow region highlights interval where cooling is evident in the temperature records.

cooling events were reported at 125.5 ± 0.5 ka and at 124.7 ± 0.9 ka[14] (Fig. 2C). Temperatures from fluid inclusion $\delta^2$H, $\delta^{13}$C, and fabric derived-$SI_{CC}$ calibrations at Cesare Battisti Cave (Trentino, Italy)[17] reveal 7.3 ± 1.9 °C, 7.1 ± 1.3 °C and 7.1 ± 1.4 °C at 126–124 ka, respectively, whereas at 124–122 ka, they decreased to 3.0 ± 1.5 °C, 5.5 ± 1.3 °C and 6.8 ± 1.2 °C (Fig. 2D). Similarly, the Grotta della Bigonda Cave in

Trentino reveals anomalous (wet/cold) events at 124.1 ± 1.8 ka, correlative with the C27 event[18]. $\delta^2$H of fluid inclusions from speleothems at Baradla Cave (Hungary), together with the $\delta^2$H/T relationship of modern rainfall, were used to reconstruct LIG air temperatures[15]. Mid-LIG cooling of up to 5 °C centered at *ca*. 124 ka was first reported; however, following a reevaluation of the $\delta^2$H/T calibration, the degree of cooling was reduced[16] (Fig. 2E−blue and red markers). The Baradla LIG temperature record notably shows similar trends to the Peqi'in $TEX_{86}$ temperature record throughout the LIG (Fig. 2F), revealing a cooling event and then a return to temperatures as high as the early-LIG. In the Eastern Mediterranean, the >3 °C mid-LIG temperature drop at Peqi'in is in good agreement with the clumped isotope temperature ($\Delta_{47}$) variations at the nearby Soreq Cave (Fig. 2F)[29].

The compiled data strongly suggests that the early-LIG was generally warmer by 3–4 °C than present during active NADW formation and relatively warm North Atlantic SST, but that temperatures decreased towards modern day values during the mid-LIG weakened AMOC interval. Importantly, the Peqi'in $TEX_{86}$ temperature record confirms and extends the known range of mid-LIG cooling associated with the weakened AMOC interval beyond Europe into the Eastern Mediterranean and Western Asia.

## Systematics of fluid inclusion stable isotopes

Fluid inclusion $\delta^{18}$O values vary between −10.0‰ and −7.3‰ (vs. VSMOW2; Supplementary Data 1). During the early-LIG from *ca*. 128–127 ka there is a decrease to −10.0‰ and then at *ca*. 126 ka values increase again to an average of −8.3‰ that lasts until *ca*. 122 ka (Fig. 3A). Compared to modern values, the fluid inclusion water $\delta^{18}$O values are lower than present-day Peqi'in Cave drip water (−5.6‰)[26] and accumulated rainfall (−6.2‰)[34], but not anomalous when compared to individual modern rainfall or moisture events in northern Israel.

Low LIG fluid inclusion water $\delta^{18}$O values at Peqi'in are typical of the LIG Eastern Mediterranean (Fig. 3B). The $\delta^{18}O_{calcite}$ at Soreq and Peqi'in generally reflects the changing $\delta^{18}$O of Eastern Mediterranean Sea surface water during the late Quaternary[26]. During the LIG, an influx of $^{18}$O depleted monsoonal-derived freshwater from North African tributaries, in particular the Nile River, led to the stratification of the seawater column and the formation of anoxic-euxinic conditions associated with deposition of organic-rich sapropel layer S5[39–41]. The $\delta^{18}O_{calcite}$ of planktic *G. ruber (albus)* from the Eastern Mediterranean Sea increased at *ca*. 126 ka. This increase was attributed to a reduced freshwater influx and a relaxation of the ITCZ and the tropical rain belt (Fig. 3C)[39,40].

Fluid inclusion water $\delta^{18}$O values at Peqi'in exhibits values and trends that are consistent with both the Soreq $\delta^{18}O_{calcite}$ and *G. ruber (albus)* $\delta^{18}O_{calcite}$ records. Nevertheless, it is important to carefully assess the possible occurrence of diagenetic alteration or other analytical or deposition artefacts[29,42]. One way to do this is to reconstruct calcite crystallization temperatures using the $\Delta\delta^{18}O$($H_2O$-calcite) method and assess if these temperatures are compatible with the $TEX_{86}$ values (Supplementary Information). Despite significant variability, the $\Delta\delta^{18}O$($H_2O$-calcite) derived temperatures generally align with, or are cooler than, the $TEX_{86}$ temperatures (Supplementary Fig. 1). The $\Delta\delta^{18}O$($H_2O$-calcite) temperatures corresponding to the early LIG low fluid inclusion $\delta^{18}$O excursion are cooler than corresponding $TEX_{86}$ temperatures. This offset may have potentially been caused by: 1) strong seasonality resulting in a decoupling between fluid inclusion water $\delta^{18}$O and $\delta^{18}O_{calcite}$[32], 2) kinetic isotope fractionation (such as prior calcite precipitation[43]) which would have increased $\delta^{18}O_{calcite}$ but not alter the fluid inclusion water $\delta^{18}$O, or 3) diagenetic alteration, such as speleothem calcite recrystallization, which caused fluid inclusion water to become $^{18}$O depleted[42]. Evidence for diagenetic alteration is weak in comparison to other mechanisms (Supplementary Information). The more realistic scenarios for the lower-than-expected

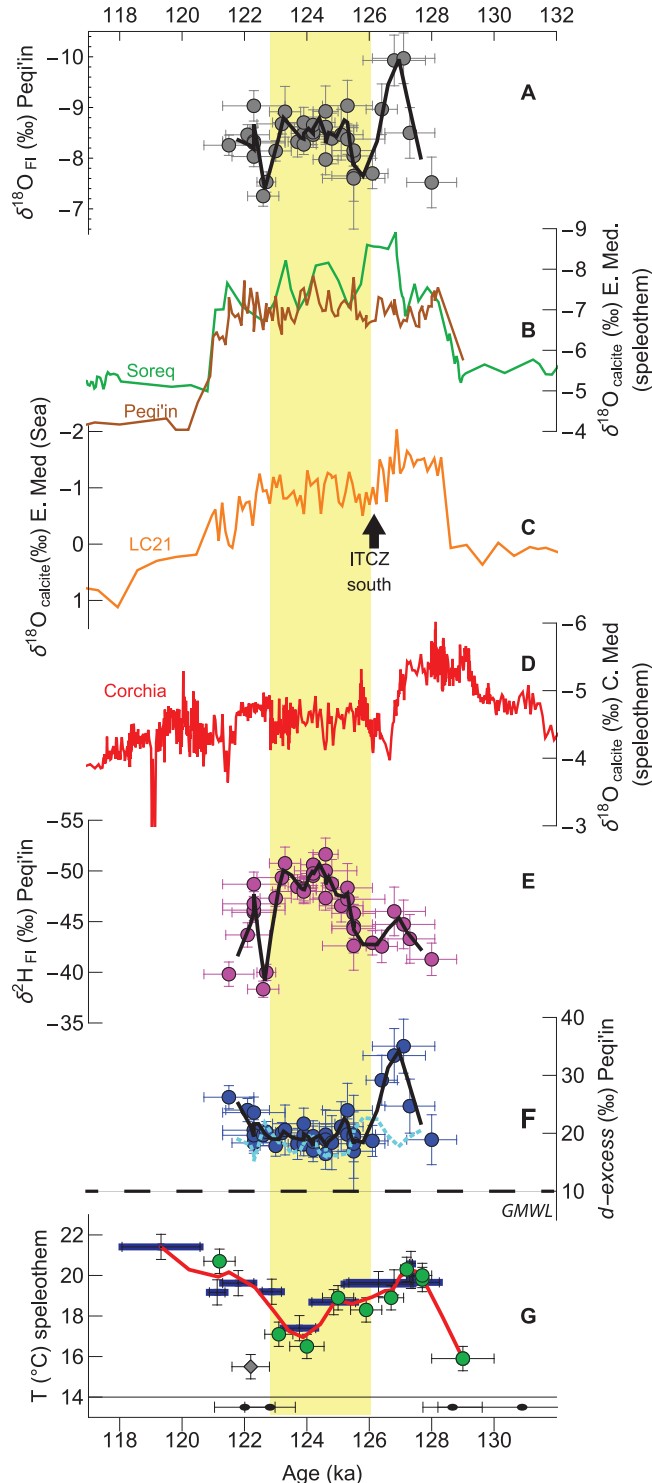

**Fig. 3 | Peqi'in Cave and Mediterranean records. A** Peqi'in fluid inclusion (FI) water $\delta^{18}O$ (grey disks) and 2 pt. median (black line). For FI isotopes, y-axis error bars represent the ±2σ of standard measurement replicates. **B** Speleothem $\delta^{18}O_{calcite}$ from Soreq (green line) and (revised) Peqi'in (brown line)[26,41]. **C** $\delta^{18}O_{calcite}$ of *Globigerinoides ruber (albus)* from Eastern Mediterranean LC21 (orange line)[41]. Black arrow illustrates when the Intertropical Convergence Zone shifted southward. **D** Speleothem $\delta^{18}O_{calcite}$ from Corchia (red line)[13,44]. **E** Peqi'in FI $\delta^2H$ (purple disks) and 2 pt. median (black line); **F** Peqi'in FI *d-excess* (blue disks), 2 pt. median (black line), and global meteoric water line (GMWL; dashed line). *d-excess$_{calc}$* calculated from FI $\delta^2H$, speleothem $\delta^{18}O_{calcite}$, $TEX_{86}$ and equation by ref. 77. (cyan dashed line). **G** Peqi'in $TEX_{86}$ temperatures from discrete samples (green disks), accumulated samples (blue rectangles), a high methane index sample (grey diamond), 2 pt. median curve (red line) and U-Th ages (lower pane; black disks; ±2σ). Yellow region highlights cooling interval.

amounts in sediments at the Portuguese margin[13], were attributed to the North Atlantic C27 event.

The Peqi'in fluid inclusion $\delta^2H$ also shows a shift from the early-LIG (−42‰ to −46‰) to the mid-LIG ~ −50‰ at *ca.* 126 ka, with a minimum at *ca.* 124 ka, before increasing to > −45‰ after *ca.* 122 ka (Fig. 3E). A similar pattern is observed in speleothems from Baradla Cave (Hungary) where fluid inclusion water shows a $\delta^2H$ minimum during the mid-LIG preceded and followed by higher $\delta^2H$ values[15,16]. At Baradla, an air temperature decrease between the early/mid-LIG was inferred to have caused the $\delta^2H$ trends.

The Peqi'in Cave fluid inclusion water *d-excess* results are in the range of 18‰ to 35‰, when directly calculated from the measured $\delta^{18}O$ and $\delta^2H$ of fluid inclusion water (Fig. 3F−blue markers), and in the range of 12‰ to 28‰, when calculated indirectly from the fluid inclusion water $\delta^2H$, $\delta^{18}O_{calcite}$ and $TEX_{86}$ temperatures (Fig. 3F−cyan line), respectively. For comparison, the *d-excess* of modern drip water at Peqi'in Cave is 23‰[25] whereas accumulated rainfall is ~20‰ (1995–2011; Fig. 1C)[34]. The Peqi'in fluid inclusion results align well with the elevated fluid inclusion *d-excess* values reported for Soreq Cave during the LIG[29] (22‰ to 29‰ calculated from measured fluid inclusion water $\delta^2H$ and $\delta^{18}O$, and 30.2 ± 8.5‰ to 42.5 ± 5.3‰ calculated using fluid inclusion water $\delta^2H$ and $\delta^{18}O_{water}$ determined using clumped isotope ($\Delta_{47}$) temperatures). The early LIG *d-excess* peaks at 35.0 ± 2.3‰, a value which is both higher than modern cave drip water and accumulated rainfall, but not anomalous for the LIG[29], nor modern moisture[45] and individual rainfall events at Peqi'in[46]. Following the early-LIG maximum, the *d-excess* decreases sharply to an average of ≤20‰ at *ca.* 126 ka and generally stays at these relatively low values until 122.5 ka. A *d-excess* minimum is found at *ca.* 124 ka (16.5‰ in direct record; 14‰ in indirect record) coincident with the maximum mid-LIG cooling as expressed by the $TEX_{86}$ temperatures (Fig. 3G). After 122 ka the fluid inclusion *d-excess* at Peqi'in increased to 26‰.

The LIG *d-excess* and $TEX_{86}$ temperature pattern at Peqi'in may be analogous to patterns seen on a glacial-interglacial timescale shown by proxies at Soreq Cave. At Soreq, the fluid inclusion *d-excess* shifts closer to GMWL values (10‰) during glacials but increases to modern Eastern Mediterranean rainfall values during interglacials[29]. To explain the glacial *d-excess* shift towards GMWL values at Soreq Cave, it was suggested that rainfall bearing storms traversed over a large zonal Mediterranean transect before reaching the Eastern Mediterranean[34], and potentially also bringing North Atlantic derived moisture[29]. However, while there is evidence of mid-LIG cooling it cannot be assumed that glacial and mid-LIG climate conditions are comparable. In the next section we investigate the controls on modern accumulated rainfall *d-excess* to address what the fluid inclusion *d-excess* changes reflect in terms of climate.

## Climatic controls of LIG d-excess values
As a critical first step for reconstructing LIG climate, the conditions which can induce *d-excess* variability in modern accumulated rainfall

temperatures are either strengthened seasonality that resulted in a decoupling between fluid inclusion water $\delta^{18}O$ and $\delta^{18}O_{calcite}$ or kinetic offsets during precipitation of speleothem $CaCO_3$. In both cases the measured fluid inclusion water $\delta^{18}O$ values are interpreted to be original but decoupled from the corresponding measured $\delta^{18}O_{calcite}$.

The increase in fluid inclusion $\delta^{18}O$ values at *ca.* 126 ka may reflect the decreased influx of freshwater runoff into the Mediterranean (ITCZ relaxation)[39,40], or the North Atlantic C27 event[13] influence on Eastern Mediterranean rainfall. A speleothem $\delta^{18}O_{calcite}$ increase up to 2 ‰ at 126.7 ka (from a $\delta^{18}O_{calcite}$ minima at 128.1 ka) at Corchia (Italy)[44] (Fig. 3D) and corresponding decrease in temperate tree pollen

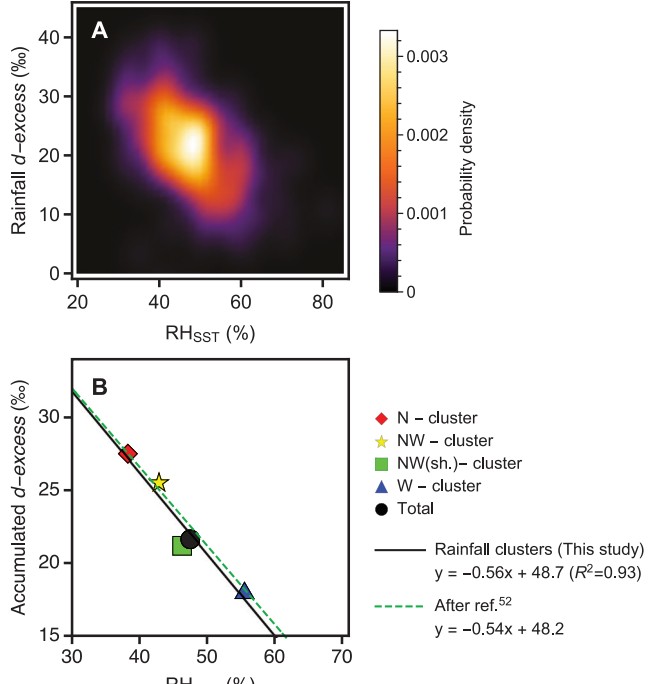

**Fig. 4 | Measured rainfall *d-excess* at Soreq vs. reanalysis derived relative humidity at 2 m calculated using sea surface temperature (RH$_{SST}$). A** Smoothed density histogram of >15 mm rainfall events. Light (dark) regions indicate high (low) data density, respectively. **B** Accumulated rainfall values of clusters (Supplementary Fig. 2; red diamond = N-cluster, yellow star = NW.-cluster; green square = NW(short)-cluster; blue triangle= W-cluster) and total accumulated rainfall (black disk). The linear regression calculated for these points is shown (black line) as is the comparable empirically-derived linear regression after ref. 52. (green dashed line).

were examined. Today, ~90% of rainfall in central and northern Israel is derived from mid-latitude Cyprus Low (CL) cyclones, where cold air mainly originating over southern Europe gathers moisture from the warm Mediterranean water before entering Israel from the west (Fig. 1B)[47–50]. The rainfall season is confined to October–May, and around two-thirds of the annual rainfall occurs during December–February[50]. CLs are synoptically classified based on both their position and their intensity[51]. The elevated *d-excess* for moisture measurements and individual rainfall events in the Eastern Mediterranean region (>15‰) reflects wintertime moisture uptake conditions, where a large negative relative humidity gradient is generated between humid air near the warm sea surface and the overlying dry air, and this results in kinetic isotopic fractionation during moisture uptake[33,45,52,53].

To determine controls of accumulated rainfall *d-excess*, we investigated the near-surface moisture uptake parameters of individual modern rainfall events that yielded >15 mm rain at Soreq sampled between 1995–2021 (previously reported: 1995 to 2003[46] to 2013[34]) by combining modern rainfall stable isotopes with backward air parcel trajectories calculated using a Lagrangian tracking tool and climate reanalysis[53]. Rainfall events were then classified (clustering) based on the direction and moisture uptake location of their associated air parcel trajectories into 4 clusters (N, NW, NWshort, W; Supplementary Fig. 2). Results from Lagrangian backward trajectory reconstruction of Soreq rainfall events reveal a positive relationship between relative humidity calculated based on SST (RH$_{SST}$) (Fig. 4) and surface air temperature (T$_{2M}$) minus sea surface temperature (SST) (T$_{2M}$ − SST) (Fig. 5). The reason for this relationship is that humidity uptake occurs in cool air relative to SST, with the result that RH$_{SST}$ is limited by the saturation vapor pressure at T$_{2M}$[53] (Supplementary Fig. 3). Despite the large *d-excess* range, a smoothed density histogram plot of rainfall

*d-excess* vs. relative humidity (RH$_{SST}$) (Fig. 4A) and T$_{2M}$ − SST (Fig. 5A) reveals that rainfall is preferably centred at ~22‰, in good accordance with the Eastern Mediterranean Meteoric Water Line[34], and corresponding moisture uptake conditions of 48% for RH$_{SST}$ and −3.5 °C for (T$_{2M}$ − SST). Replotting the accumulated rainfall values for the clusters reveals the influence of moisture uptake location in the Mediterranean on rainfall *d-excess* (Figs. 4B and 5B). Two extreme *d-excess* cluster endmembers were produced (Supplementary Fig. 2; Fig.5C inset): (i) meridional N-cluster with moisture uptake exclusively in the Eastern Mediterranean (Aegean Sea and Levantine Sea) and high accumulated *d-excess* values (27.5‰), and (ii) zonal W-cluster with moisture source uptake zonally across the Mediterranean Sea (where SSTs are lower; Fig. 1B), a small fraction of North Atlantic moisture, and lower *d-excess* values (18.1‰). The cluster *d-excess* values have associated RH$_{SST}$ values that, together with the total accumulated rainfall at Soreq, exhibit a robust negative linear correlation (Fig. 4B; R$^2$ = 0.933) and a striking similarity with the linear regression calculated from compiled moisture and rainfall events after ref. 52. Important for understanding the LIG record, the clusters also exhibit a robust negative linear correlation with low (high) T$_{2M}$ − SST and high (low) *d-excess* (Fig. 5B). The 1995–2021 total accumulated rainfall *d-excess* value (21.6‰) illustrates the contribution of rainfall derived from the clusters at varying proportions. Therefore, over a given period (seasons, years, etc.), atmospheric circulation shifts that shift the location of moisture uptake source will also change the accumulated rainfall *d-excess* and the local Eastern Mediterranean Meteoric Water line. These findings justify the use of speleothem fluid inclusion *d-excess* records as both a measure of cumulative moisture source conditions and atmospheric circulation.

What caused the LIG fluid inclusion *d-excess* values and trends at Peqi'in? A debated atmospheric circulation feature suggested to cause the ¹⁸O depleted Eastern Mediterranean speleothem records is the northward shift of the Intertropical Convergence Zone (ITCZ) and the tropical rain belt over North Africa and into the Levant, leading to a direct contribution of summer monsoonal rainfall[54]. However, these studies did not consider a mid-LIG weakened AMOC, which is associated with a southward shift of the ITCZ and the tropical rain belt[55]. The range of fluid inclusion *d-excess* values at Peqi'in, together with the T$_{2M}$ − SST and *d-excess* relationship, provide evidence that the LIG rainfall source was always the adjacent Mediterranean Sea and contradicts the summer monsoon intrusion hypothesis. It is notable that in southern Arabia at Mukallah Cave (Yemen), a site which partly receives summer monsoon rainfall today (Fig. 1A), speleothem fluid inclusion water from the LIG exhibits isotopic values close to the modern GMWL[56,57].

An absolute comparison of modern rainfall cluster *d-excess* values and LIG *d-excess* values cannot be made because the Earth's orbital configuration during the LIG was markedly different from today and this resulted in increased summer-winter insolation (and temperature) differences (Supplementary Fig. 1). Strong seasonal temperature differences likely amplified air-sea temperature gradients in the wintertime Mediterranean Sea. Considering the robust modern T$_{2M}$ − SST and *d-excess* relationship, and the increased seasonality, one would expect the LIG *d-excess* to have been higher than today. Indeed, values higher than modern *d-excess* (~35‰) are found near the precession minimum (*ca.* 127 ka), a period of maximum summer insolation and minimum winter insolation, and, additionally, a period of reduced AABW but strengthened NADW formation[9,10] (i.e., strengthened AMOC). However, at *ca.* 126 ka the fluid inclusion *d-excess* decreases to <20‰ and generally persists until *ca.* 122 ka, with a minimum at *ca.* 124 ka. To explain this decrease by changes at a fixed moisture uptake source and no change in air temperature requires a large SST cooling by >4–6 °C, which is not significantly evident in Eastern Mediterranean Sea SST proxy records[58]. Therefore, the abrupt *d-excess* decrease at 126 ka most likely reflects an atmospheric circulation reorganization in which rainfall was derived from increased

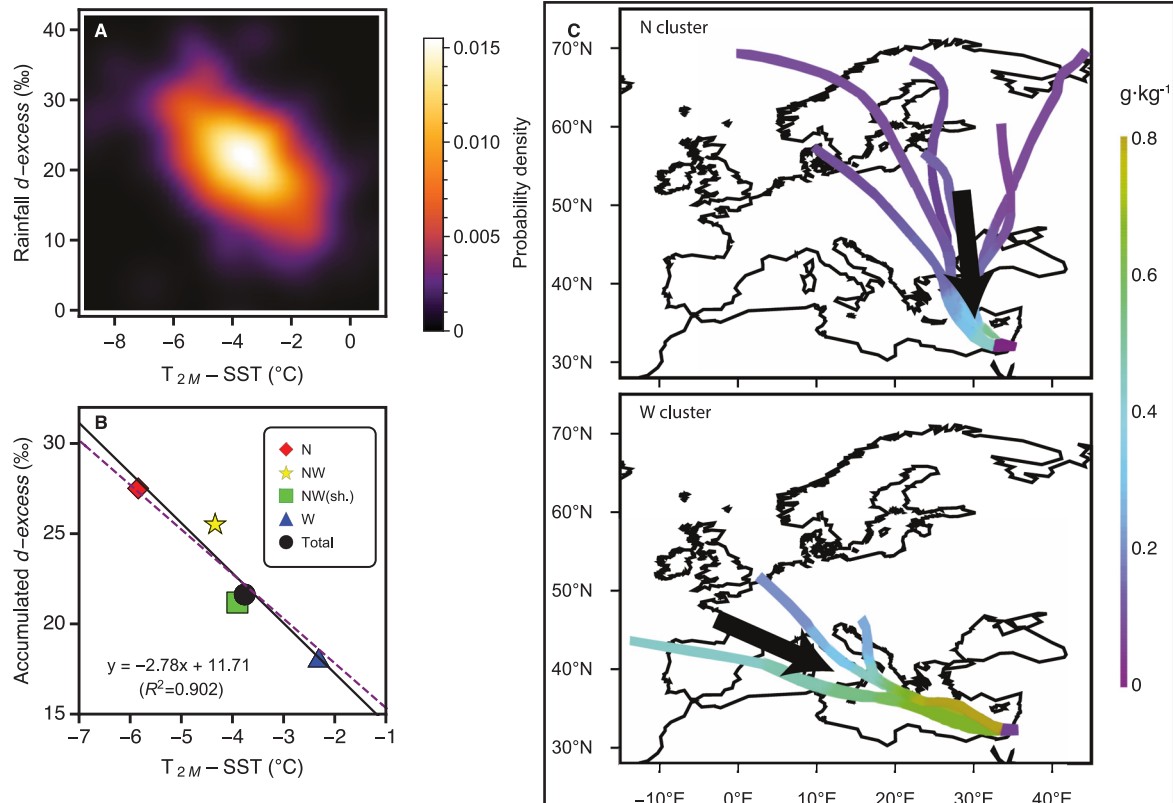

**Fig. 5 | Measured rainfall *d-excess* at Soreq vs. reanalysis derived 2 m air temperature minus sea surface temperature ($T_{2M}$ − SST). A** Smoothed density histogram of >15 mm rainfall events. Light (dark) regions indicate high (low) data density, respectively. **B** Accumulated rainfall values of clusters (Supplementary Fig. 2; red diamond= N-cluster, yellow star = NW.-cluster; green square=NW(short)-cluster; blue triangle= W-cluster) and total accumulated rainfall (black disk). Additionally shown are the linear regression of these points (black line and equation) and the linear regression of near surface moisture measurements at Rehovot, Israel after ref. [53]. (purple dashed line). **C** The two extreme *d-excess* clusters (meridional N-cluster and zonal W-cluster) and their characteristic trajectories. Specific humidity changes are illustrated by colour changes along the trajectories and emphasize the location of dominant moisture uptake.

moisture uptake in the central and, possibly, western Mediterranean Sea, where SSTs were cooler (Fig. 1B) (i.e. an increased proportion of air parcels originating in the North Atlantic, which traversed zonally over the Mediterranean Sea to provide Eastern Mediterranean rainfall). Furthermore, based on modern cluster rainfall season distributions, the mid-LIG atmospheric circulation change may have been accompanied by changes in rainfall season length, with more rainfall occurring at the ends of the rainfall season (Supplementary Fig. 4). Given the evidence of North Atlantic SST cooling during that time (Fig. 2B), increased air flow from the North Atlantic to the Eastern Mediterranean may have provided an atmospheric mechanism facilitating Eastern Mediterranean cooling between *ca.* 126–123 ka.

The physical mechanism linking mid-LIG changes in wind direction over the Eastern Mediterranean can be further investigated using results from LIG climate model experiments simulating a weakened AMOC. In the study by ref. [13], North Atlantic freshwater discharge simulations were made using Earth System model LOVECLIM44. These simulations reveal strengthened westerlies over the North Atlantic that appear to penetrate into the Western Mediterranean. This raises the question whether North Atlantic westerlies continued their eastward trajectory to facilitate Eastern Mediterranean rainfall. Under strengthened meltwater influx (0.05 Sv) to the North Atlantic, the simulations reveal a positive geopotential height (500hPa) anomaly (annual mean changes relative to a control LIG state) at the far-west Mediterranean (Straits of Gibraltar) and a negative anomaly in the Eastern Mediterranean. Specifically for the Eastern Mediterranean region, wind vector anomalies in these models show a SWW direction component. The elevated *d-excess* values prior to *ca.* 126 ka and following *ca.* 122 ka suggest moisture uptake localized in the Eastern

Mediterranean (Aegean Sea and Levantine Sea) via meridional northly air flow over central Europe. This observation fits with the control LIG state in the experiments by ref. [13]. characterized by a strengthened AMOC and NADW formation.

## AMOC related rainfall source changes in recent years?

Although the fluid inclusion *d-excess* record shows abrupt changes on a centennial scale, the rate of atmospheric reorganization cannot be accurately estimated. To help better constrain the use of the fluid inclusion *d-excess* as a proxy, we investigate if large accumulated rainfall *d-excess* changes can occur on multi-annual and decadal timescales. As we will show this has implications for the use of rainfall *d-excess* in studying shifts in the modern rainfall regime.

Modern accumulated rainfall from numerous monitoring sites in recent years exhibit large *d-excess* changes from 24–26‰ during 2002–2004 to 14–18‰ during 2017–2019 (Fig. 6A). Based on the strong ($T_{2M}$ − SST) and *d-excess* relationship (Fig. 5B), the recent decreasing *d-excess* trend can be assumed to be the result of increasing wintertime air temperatures in the Mediterranean region and an overall reduced air-sea (surface) temperature gradient during moisture uptake[59]. However, there are also signs of atmospheric circulation changes associated with the reduced accumulated rainfall *d-excess*. An example is the accumulated rainfall of the 2017–2019 rainfall seasons characterized by particularly low *d-excess* (14/16‰) which is also represented by an increased frequency of North Atlantic derived air flow (Fig. 6A), although it is important to note that there is no absolute increase in the frequency of events for W-cluster alone.

Given that >90% of rainfall is derived from CL's, raises the question of whether CL tracks have repositioned over the Mediterranean in

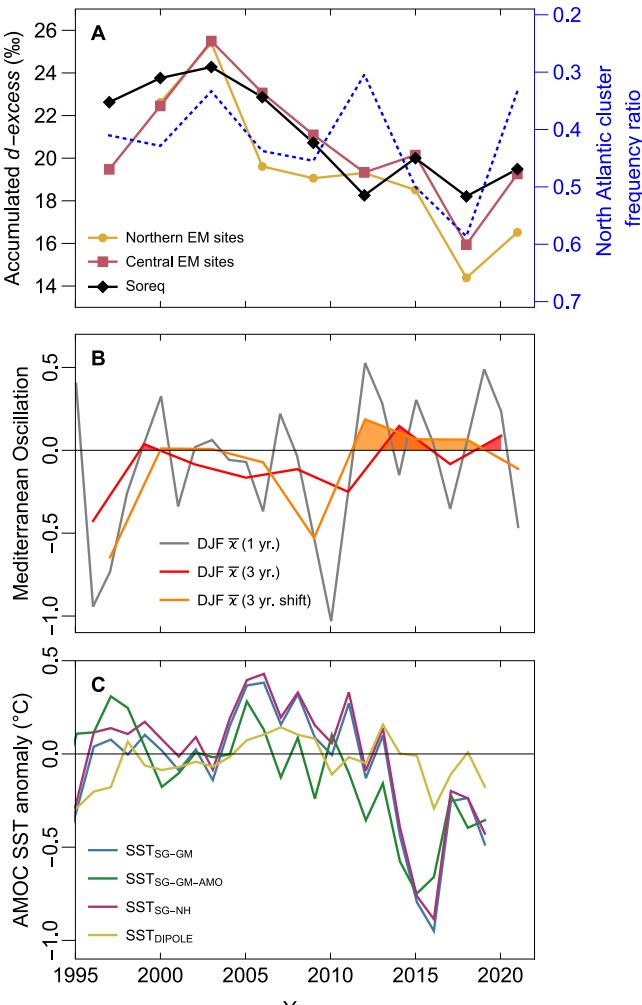

**Fig. 6 | Recent Eastern Mediterranean rainfall *d-excess*, Mediterranean Oscillation Index and Atlantic Meridional Overturning Circulation index trends.**
**A** Accumulated rainfall *d-excess* values calculated over three-year intervals for northern rainfall collection sites (yellow disks and line), central rainfall collection sites (red squares and line), and Soreq (black diamonds and line). Superimposed on the plot is the North Atlantic cluster frequency ratio derived from rainfall at Soreq (blue dashed line). **B** The December to February (DJF) annual mean (grey line) and three-year mean (red and orange lines) of the Mediterranean Oscillation Index (MOI$_2$). A positive MOI$_2$ was shown to correlate with a southward shift of the CL tracks reaching Israel[21,24]. **C** Atlantic meridional overturning circulation (AMOC) indices based on mean sea surface temperature (SST) differences[3,5]. Negative (positive) trends infer a weakening (strengthening) AMOC. SST$_{SG-GM}$ (blue line): defined as the difference between the mean SST of the subpolar gyre region (SG) and the whole globe. SST$_{SG-GM-AMO}$ (green line): based on the same spatial regions as SST$_{SG-GM}$ but with the contribution of the Atlantic Multidecadal Oscillation (AMO) removed. SST$_{SG-NH}$ (purple line): defined as the difference between the mean SST of SG and the Northern Hemisphere. SST$_{DIPOLE}$ (yellow line): defined as the difference between South-Atlantic and North-Atlantic SST's. These indices correlate well with actual AMOC strength in simulations with freshwater hosing and gradual CO$_2$ increase models.

recent years and contributed to the decrease in accumulated rainfall *d-excess*. Recent studies suggest a teleconnection between CL track position and atmospheric climate indices. An important climate index for this region is the Mediterranean Oscillation Index (MOI$_2$; or Mediterranean Pressure Index)[21], defined as the upper-level pressure difference between the Western Mediterranean (Gibraltar) and Eastern Mediterranean (Lod, Israel) (Fig. 1A)[60]. Spatial assessment revealed that a positive MOI$_2$ correlates well with a southward shift of CL track position and rainfall in Israel[24]. The December-February averages

of MOI$_2$ indeed reveal positive values following 2010 (Fig. 6B); trends which support the reduced accumulated rainfall *d-excess* and increased frequency of rainfall via North Atlantic derived air parcels (Fig. 6A).

The Eastern Mediterranean rainfall and atmospheric trends occur against the backdrop of a weakened AMOC (Fig. 6C)[3,5], and reports of reductions in the total rainfall amounts in Israel[25]. While the contribution of CL track reorganization vs. differential rates of air / SST warming on rainfall *d-excess* needs to be better quantified, the large *d-excess* decrease in recent years illustrates how the Eastern Mediterranean Meteoric Water Line is not stationary and has been rapidly shifting in this region. Is this evidence of an atmospheric reorganization in the Eastern Mediterranean occurring with a weakening AMOC? Further investigation into how the Eastern Mediterranean rainfall regime and the MOI$_2$ respond to an AMOC slowdown will be key to assessing a teleconnection. However, if the answer to this question is yes, then the mid-LIG climate may serve as a good analogy for what may lie ahead.

## Methods
### Peqi'in and Soreq caves and speleothems
The speleothems used in this study were taken from Peqi'in Cave and Soreq Cave and investigated in earlier studies[26–29,61–65]. Peqi'in Cave, located in northern Israel (35.329°N, 32.976°E), has a length of 17 m, a width of 4.5–8 m, and is about 10–30 m below the surface within Turonian (Late Cretaceous; 93.9–89.8 million years ago) limestone host-rock. It consists of three levels sloping down from east to west[66]. The cave was used by humans during the Late Chalcolithic period (between 4500–3900 BCE) and served as a burial site[67]. Subsequent to *ca.* 3900 BCE, it was sealed by natural processes (collapse of the entrance, most likely by earthquake) before being discovered during road construction in 1995. The cave contains stalactite and stalagmites producing late Quaternary paleoclimate records[26].

Soreq Cave is located in central Israel (31.756°N, 35.023°E), on the western slopes of the Judean Hills, south of Nahal Soreq stream. Its maximum dimensions are 90 m long, 80 m wide, and 15 m high and is about 10–50 m below the surface within Cenomanian (100.5–93.9 million years ago) dolomitic host-rock. Today it is open to visitors and the cave environment and climate are monitored by the Israel Nature and Parks Authority. Reported mean annual temperatures were 16 °C for Peqi'in and 18/19 °C for Soreq; values that align with temperatures calculated from the δ18O difference between modern soda straw stalactites (calcite) and average drip water isotope composition[26].

We used Peqi'in PEK-9 (stalactite) for fluid inclusion stable isotope and TEX$_{86}$ temperature reconstruction[26]. We also studied three Soreq speleothems: SO-15 (flow stone), 12-Z and 2-6 (stalagmites) for TEX$_{86}$ temperature variations between the last interglacial period (LIG)[62,63], the last glacial maximum[61], and the late Holocene[64], respectively. PEK-9 was sampled along its growth axis perpendicular to the lateral and radial growth lamina. Early studies originally dated these speleothems using $^{230}$Th-$^{234}$U method on thermal ionization mass spectrometry (TIMS)[26,61,65] (details of ages for PEK-9 for the laminae relevant in this study can be found in Supplementary Table 1). In a later study at Soreq, high-precision dating was carried out using multi-collector inductively coupled plasma mass spectrometry (MC-ICP-MS)[41]. A paired coherency was reported between U/Th ages acquired originally by TIMS and MC-ICP-MS producing virtually identical age models for the Soreq δ18O$_{calcite}$ record (Supplementary material in ref. 41). Given this MC-ICP-MS and TIMS dating coherency at Soreq, we thus used the original TIMS-derived ages for PEK-9 determined by ref. 26,61. but revised the PEK-9 age model by aligning the LIG Peqi'in δ18O$_{calcite}$ to the revised Soreq δ18O$_{calcite}$ record[41] within the TIMS derived absolute ages ±2σ (Supplementary Fig. 5). Based on these absolute age uncertainties (±2σ), the age uncertainties are ±1 ka for nearly all samples. The

temporal uncertainties (±) for fluid inclusion and TEX$_{86}$ samples reported (Supplementary Data 1) reflect the sample size derived uncertainties.

In this study we define the early-LIG from 132 ka, the transition from Marine Isotope Stage (MIS) 6 to MIS 5e, until 126 ka, following precession minima and maximum/minimum summer/winter insolation, and the mid-LIG here is defined as starting at 126 ka.

Polished and covered thin sections of PEK-9 were produced to investigate the fabric and distribution of fluid inclusions. The thin sections were observed and photographed under plane polar light (PPL) and crossed polar light (XPL) on a Zeiss Axiophot microscope. Crystalline and fluid inclusion features, size and distribution were documented. The PEK-9 thin sections reveal columnar fabric where crystals are distributed perpendicular to the growth axis, slightly radial and typical of syntaxial stalactites. Within the crystals are elongated μm and sub-μm scale thorn shaped fluid inclusions with orientation parallel to the crystal optic axes (Supplementary Fig. 6). Some faint layers exist in the fabric where fluid inclusions are sometime more frequently distributed.

To detect non-CaCO$_3$ impurities, an additional non-covered thin section was prepared and analysed for surface elemental ratios using a Leo 1530 Field Emission Scanning Electron Microscope (SEM) with spot surface chemical fraction measurements using energy dispersive x-ray analysis (EDX) X-MAX 80 from OXFORD instruments (SEM-EDX) at the Max-Planck Institute for Chemistry.

## Fluid inclusion water stable isotope analysis

Speleothem fluid inclusion water and calcite stable isotopes were measured at the Inorganic Gas Isotope Geochemistry laboratory at the Max-Planck Institute for Chemistry. PEK-9 was sampled for fluid inclusions in batches. Calcite chips (0.2–1.3 gr) were cut using a Dremel Microdrill equipped with a 38 mm diameter diamond-covered circular saw along and parallel to the lateral growth axis (longest lateral section; maximum offset from axis of ±18 mm). Off-axis samples were extrapolated along the layer curvature to the axis and sample ages were given respective to their position along the axis. The first sampling batch was used to determine the spatial distribution of fluid inclusion stable isotope variability in the speleothem laminae for the entire LIG. Further sampling of the LIG section was done to determine reproducibility.

Fluid inclusion water stable isotopes were measured using a humidified (H$_2$O) nitrogen (N$_2$) carrier gas line at 110 °C, connected via a crusher unit to a Picarro L2140i cavity ring-down spectrometer (CRDS) where H$_2$O concentrations, $\delta^2$H and $\delta^{18}$O were continuously measured[68,69]. The system was shown to have insignificant memory effect and minimal long-term instrumental drift[70]. Speleothem chips were crushed in a stainless-steel crusher unit with two Teflon coated Viton O-Rings seals by a downward rotary movement which released fluid inclusion water directly mixed into the humidified carrier gas line for subsequent isotope analysis. Following measurement, the $\delta^2$H and $\delta^{18}$O of samples (injected standard waters and fluid inclusion water) were calculated using a python script according to ref. 68. First, mean pre-peak and post-peak integrated background vapor isotope values and water concentrations were calculated, and then subtracted from the respective whole integrated sample (+ background) peak water isotope value and concentrations. Further details of the data processing methods and calibration can be found in ref. 69.

Water vapor concentrations and stable isotope compositions of the background vapor were recorded prior to and following sample crushing. These pre-peak and post-peak background vapor values ideally should be identical and thus were used as criteria for detecting erroneous samples. Analytical precisions are dependent on sample water volumes as well as sample isotope proximity to background isotope water values (Background vapor values: $\delta^{18}$O = −8.70‰ and $\delta^2$H = −62.96‰; vs. VSMOW)[69,70]. Based on near-equivalent isotope

composition standards (relative to fluid inclusion values) the measurement uncertainty (1σ) for the binned H$_2$O yields of 0.08–0.15 μL, 0.15–0.25 μL, 0.25–0.80 μL are: 0.25‰, 0.15‰, & 0.10‰ for $\delta^{18}$O, respectively. For the same H$_2$O yield bins, $\delta^2$H measurement uncertainties are 1.2‰, 0.8‰, 0.6‰ & 0.4‰, respectively. The measurement uncertainty for *d-excess* is between 2.3‰ and 0.9‰ (1σ) based on propagation of error. Significantly reduced analytical precision occurs below the 0.08 μL threshold[69], the minimum water yield volume for calibration. Results are given in permille (‰) vs. VSMOW. Following each crush, the post-crush CaCO$_3$ material was removed and both the sample holder and piston of the crusher unit were cleaned with acetic acid, rinsed thoroughly with milliQ water and dried with pressurized air. The sample holder was loaded with a new sample and placed into the oven (i.e. not connected online to the CRDS), allowing for equilibration to the oven temperature and any adsorbed water on the sample chips to evaporate.

In most cases, fluid inclusion H$_2$O yields were adequate for stable isotope analysis and sample batches exhibit robust reproducibility. However, there were some samples with exceptionally low water yield that could not be calibrated (Supplementary Data 1). Additionally, for samples dating from *ca*. 118 ka to 122 ka, the isotopic composition of background vapor and the injected standard were instable and fluid inclusion isotopes could not be measured. SEM-EDX analysis of thin section detected Al, Si, K, and Fe elements in those respective speleothem laminae. This suggests the presence of clays and/or oxides/hydroxides which are likely to have interfered with fluid inclusion isotope measurement.

Post-crush material aliquots of 50–200 μg were prepared for $\delta^{13}$C$_{calcite}$ and $\delta^{18}$O$_{calcite}$ analyses. $\delta^{13}$C$_{calcite}$ and $\delta^{18}$O$_{calcite}$ were measured on a Thermo Delta V IRMS fitted with a GASBENCH. Cleaned 12 ml vials containing the calcite were initially flushed with Helium followed by injection of H$_3$PO$_4$ and subsequent calcite dissolution (at 70 °C). The CO$_2$–He mixture was then transferred to the GASBENCH for water vapor and residual gaseous compound separation before isotope measurement. Isotope values are reported as $\delta^{13}$C$_{calcite}$ and $\delta^{18}$O$_{calcite}$ in ‰ relative to VPDB. Standard reproducibility (1σ) is better than 0.1 ‰ for both $\delta^{13}$C$_{calcite}$ and $\delta^{18}$O$_{calcite}$. Supplementary Fig. 7 shows the $\delta^{13}$C$_{calcite}$ and $\delta^{18}$O$_{calcite}$ timeseries records of Peqi'in together with the respective post-crush material (and TEX$_{86}$ sample) aliquot isotope values. Generally, samples taken off-axis but in the same laminae have comparable calcite $\delta^{18}$O$_{calcite}$ and $\delta^{13}$C$_{calcite}$.

## GDGT (TEX86) analysis

Glycerol Dialkyl Glycerol Tetraethers (GDGTs) are cell membrane lipids produced by either archaea or bacteria and are widespread in both marine and terrestrial environments[71]. The distribution of GDGT lipids in speleothems can be used to estimate and reconstruct cave paleo-temperatures using the TEX$_{86}$ index[32,72,73]. GDGT's were measured at the Organic Isotope Geochemistry laboratory at the Max-Planck Institute for Chemistry. For PEK-9, two speleothem sample sets were measured: a first set of accumulated post-crush sample material from adjacent samples and a second set of non-crushed sample batch taken discretely along the growth axis.

Samples were placed in a 60 mL cylindrical glass vial before being digested in 5-20 mL (~5 mL per gram of sample) of 6 N HCl on a heating plate at 100 °C for 2 h. The dissolved samples were then cooled to room temperature and extracted in a separation funnel using 3 × 20 mL dichloromethane (DCM). The bottom organic-DCM phases were carefully collected in 60 ml vials, and 40 μL of C$_{46}$ internal standard was added to the extracts. Extracts were evaporated and dried in an Automated Evaporation System (Rocket Evaporator), followed by re-dissolution in a DCM–methanol mixture. Traces of acid and other impurities were removed by passing solutions through a 5% deactivated silica column and collected in 4 mL vials. The eluates were dried and filtered with a 0.2 μm PTFE membrane filter by rinsing with a 1.8%

mixture of isopropanol in n-hexane. Measurements of the extracts were done using an Agilent 1260 Infinity high-performance liquid chromatography (HPLC) coupled to an Agilent 6130 single-quadrupole MS[74]. GDGT signals for the isoprenoid GDGT compounds (GDGT-0, GDGT-1, GDGT-2, GDGT-3, crenarchaeol, crenarchaeol isomer (Cren')) and GDGT $C_{46}$ were integrated using the Agilent LC/GC-MS data analysis software Mass Hunter according to their mass-to-charge ratio (m/z). The tetraether index (TEX$_{86}$) was calculated using the following equation (Eq. 1):

$$TEX_{86} = \frac{GDGT2 + GDGT3 + Cren'}{GDGT1 + GDGT2 + GDGT3 + Cren'} \quad (1)$$

We used the following calibration provided by ref. 19. to reconstruct cave temperatures (T$_{cave}$; unscreened data) (Eq. 2):

$$T_{cave} = 7.34(\pm 2.71) + 34.64(\pm 4.16) \cdot TEX_{86} \quad (2)$$

A homogenized flowstone speleothem from Scladina Cave (Belgium) was used as an internal standard for every GDGT analysis batch[32]. In addition, an extract of the Scladina standard was measured at various dilutions to determine the minimum concentration threshold for temperature calibration and precision. Scladina standard measurements produce an average cave T of 14.74 °C ± 0.31 °C (1σ; $n = 28$) which compares well to reported value in an earlier study by ref. 32. Y-axis error bars in the figures (Figs. 2F and 3G) are based on the external analytical precision (2σ) derived from repeated measurements of the speleothem standard (i.e. 0.62 °C).

Quality checks were made for potential influencing factors on GDGT distributions which can alter TEX$_{86}$. In marine and lake sediments, GDGTs produced by methanogens can bias TEX$_{86}$ temperatures estimates[72]. To distinguish between normal marine and gas hydrate impacted and/or methane-rich environments the methane index (MI) was suggested[75] (Eq. 3):

$$MI = \frac{GDGT1 + GDGT2 + GDGT3}{GDGT1 + GDGT2 + GDGT3 + Cren + Cren'} \quad (3)$$

Additionally, we assessed whether methanogenic archaea may have influenced GDGT distributions using the calculation of the GDGT-0/Crenarchaeol[76]. It was suggested that a threshold of GDGT-0/Crenarchaeol >2 may be indicative of a methanogenic source of GDGT-0 (assuming that methanogens produce GDGT-0 but no Crenarchaeol). As of yet it is not clear if methanogens in the karstic environment may affect the GDGT distribution and subsequent temperature reconstructions, but its potential influence cannot be ignored (Supplementary Fig. 8).

Measured temperatures following opening at Peqi'in and Soreq differed by 2–3 °C (16 °C for Peqi'in and 18/19 °C for Soreq)[26]. The corresponding LIG Soreq speleothem (SO-15) generally produced too low GDGT concentrations for TEX$_{86}$ calibration, apart from three samples. Soreq LIG samples produced 21.6 °C (1σ = 0.3 °C; $n = 3$). These LIG TEX$_{86}$ temperatures are within the upper range of LIG clumped isotope (Δ$_{47}$) temperatures, excluding the samples showing mid-LIG cooling[29].

The TEX$_{86}$ temperatures at Peqi'in align with the values and trends of clumped isotope (Δ$_{47}$) temperatures from Soreq (not only for the LIG; see within manuscript) corroborating the use of both methods. Samples from the Soreq Cave 12-Z speleothem dating to the last glacial maximum yielded some exceptionally low GDGT concentrations, but for the rest, TEX$_{86}$ temperatures measured an average of 12.6 °C (1σ = 1.1 °C; $n = 3$) which compares well with 11.6 ± 0.6 °C for Δ$_{47}$ for speleothem 12-Z at ca. 20 ka. The Soreq 2-6 speleothem dating to the late Holocene (ca. 1.3 ka) yielded a TEX$_{86}$ temperature of 17.2 °C

(1σ = 1.5 °C; $n = 5$) and within error for respective Δ$_{47}$ values reported for this period of 19.0 ± 1.5 °C for Δ$_{47}$.

In addition to the TEX$_{86}$ temperatures, we reconstructed cave temperatures using fluid inclusion water δ$^{18}$O and speleothem δ$^{18}$O$_{calcite}$ (Δδ$^{18}$O method; Supplementary Information). Unlike the TEX$_{86}$ temperatures, temperatures derived from the Δδ$^{18}$O during the early-LIG at Peqi'in are lower than expected. Although it is unlikely that fluid inclusion water δ$^{18}$O was altered diagenetically (Supplementary Information) it cannot be completely ruled out. To address the potential effects of altered fluid inclusion water δ$^{18}$O on d-excess, we calculated an alternate d-excess record (d-excess$_{calc.}$; Fig. 3F−cyan dashed line; Supplementary Data 1) using the fluid inclusion water δ$^2$H, δ$^{18}$O$_{calcite}$, interpolated TEX$_{86}$ temperatures and implementing the empirical equation by ref. 77. which was shown to conform well to the Peqi'in TEX$_{86}$ temperatures during the mid-LIG (Supplementary Information) (Eq. 4):

$$d - excess_{calc.} (‰) = \delta^{18}H_{2}O - 8 \cdot \left( \frac{\delta^{18}O_{calcite} + 10^3}{e^{(17.57 \times 10^3 / T_{TEX_{86}} - 29.13)/10^3} - 10^3} \right) \quad (4)$$

Where δ$^{18}$O$_{calcite}$ is in ‰ vs. VSMOW (converted using: δ$^{18}$O$_{VSMOW}$ = δ$^{18}$O$_{VPDB}$ × 1.03086 + 30.86) and T$_{TEX_{86}}$ are the linear-interpolated age-equivalent TEX$_{86}$ temperatures in kelvin. Although an early-LIG maximum is not evident in d-excess$_{calc.}$ (Fig. 3F) we find a minimum at ca. 124 ka which supports our interpretation of the fluid inclusion d-excess record.

## Rainfall collection and analysis

Rainfall collection for stable isotope analysis was carried out at multiple sites in Israel (Fig. 1C; Supplementary Table 2)[34,46,78]. In this study we use rainfall stable isotope data collected above Soreq Cave between the years of 1995 to 2021 (last rainfall event dated: 26/03/2021) for Lagrangian backward air parcel trajectory calculations. A total of 869 rainfall collections and measurements were made, carried out by park rangers operating the Soreq Cave Nature Reserve and researchers at the Geological Survey of Israel. Rainfall was collected by allowing the water to accumulate in a large funnel dripping into a narrow-headed bottle to reduce evaporation effects. Events were sampled with a daily resolution during the rainfall season. During some intensive rainfall events, the collecting bottles were changed during the day several times to minimize water loss. Accumulated daily rainfall and the respective stable isotopes were calculated for days where more than one rainfall collection bottle was used. The rainfall amount was calculated by dividing the amount of water that was accumulated in the collecting bottle by the surface area of the funnel.

For rainfall δ$^{18}$O measurements, clean vacuum vessels were flushed with a gas mixture of He (99.6%) and CO$_2$ (0.4%) for 10 minutes. After flushing, 0.7 cm of the sampled water was injected to the vessels and left to equilibrate with the CO$_2$ gas for at least 48 hours at 25 °C. Values of δ$^{18}$O were measured using a Finnigan GASBENCH II extraction system in continuous flow connection with a Thermo Finnigan Delta PLUS XP mass-spectrometer at the Geological Survey of Israel (GSI), following the CO$_2$ −H$_2$O equilibration technique. All oxygen isotopic measurements were made in duplicates and are reported relative to VSMOW. Four well-quantified internal laboratory standards were used for calibration, and a standard was measured after every eight water samples. Values of δ$^2$H were measured at the GSI using a Thermo Finnigan High Temperature Conversion Elemental Analyzer (TC-EA) attached to a Delta V mass-spectrometer at a reaction temperature of 1450 °C. Each sample was measured 6 times, and δ$^2$H values are reported relative to VSMOW. Measurement uncertainty based on replicate measurements was better than 1.0‰ for δ$^2$H and 0.1‰ for δ$^{18}$O.

Rainfall *d-excess* was calculated by (Eq. 5):

$$d - excess = \delta^2 H - 8 \cdot \delta^{18} O \qquad (5)$$

Based on error propagation the measurement uncertainty of *d-excess* is 1.3‰. Accumulated rainfall *d-excess* was calculated by (Eq. 6):

$$\text{Accumulated } d - excess = \frac{\sum d - excess \times \text{rainfall}}{\sum \text{rainfall}} \qquad (6)$$

Where rainfall is the amount of water per event in mm. Backward trajectories were calculated for events with rainfall >15 mm as events with lower water yields were suggested to have been isotopically altered by evaporation beneath the clouds[34,46,76]. The removal of these events is justified as the contribution of <15 mm rainfall events to the total accumulated rainfall *d-excess* at Soreq from 1995–2021 is minor (21.4‰ for >15 mm events; 20.6‰ for all events). In total, backward trajectories were calculated for 263 rainfall days following the removal of <15 mm rainfall events.

Accumulated rainfall *d-excess* for three-year intervals at Soreq, central sites and northern sites were calculated (Fig. 6A). Central and northern sites were grouped according to their proximity and whether they had accumulated *d-excess* values typical of the Eastern Mediterranean meteoric water line[34] (Fig. 1C; Supplementary Table 2).

## Lagrangian moisture source diagnostics and clustering

The conditions at the near surface air-water interface during moisture uptake reconstructed using climate reanalysis models were shown in earlier studies to be associated with measured water vapor *d-excess* at Rehovot, Israel[53,79]. Critical for understanding modern and paleo multi-annual *d-excess* changes, we investigated whether the same relationships hold for both individual rainfall events and accumulated long-term rainfall. For this, backward trajectories of daily rainfall events measured at Soreq were calculated with the Lagrangian analysis tool, LAGRANTO (v2.0)[80] using 3-dimensional wind fields from the fifth-generation reanalysis produced by ECMWF (ERA5)[81]. The ERA5 variables were taken with a 3-hourly temporal resolution, with a horizontal resolution of 0.5° × 0.5°, and 131 vertical levels from the surface up to 0.01 hPa. Additional variables from ERA5 that were used include specific ($q$) and relative humidity ($RH_{SST}$), 2-meter temperature ($T_{2M}$), sea surface temperature (SST), atmospheric and surface pressure (PS), sea-level pressure (SLP), and boundary layer height (BLH). Air parcel trajectories which arrived at the lower troposphere (below 600 hPa) at the selected days of rainfall events are considered by calculating back-trajectories for every odd model level of the lowest 37 levels (1,3,...37). Trajectories were calculated 10 days back at 3-hour intervals for each measurement date starting at time 00 UTC. The moisture uptake, defined as an increase in specific humidity between two consecutive time steps along the trajectories, was evaluated, but required two criteria to be initially tested: 1. The two-time steps average of the trajectory pressure is higher than the boundary layer pressure (BLP) defined by (Eq. 7):

$$BLP = P_0 \cdot e^{-\frac{1.5 \cdot BLH}{8000}} \qquad (7)$$

Where $P_0$ is the standard atmosphere and the factor of 1.5 accounts for parametrization uncertainties of the BLH and high moisture uptake close to the top of the boundary layer[53,79]. 2. The trajectory is located over a marine source (for at least one of the two-time steps), identified with finite SST values.

The relative contribution of each moisture uptake to the final humidity was calculated ($f$; a fraction between 0 and 1). To remove potential numerical uncertainty only humidity changes >0.01 g·kg⁻¹ (or <−0.01 g·kg⁻¹, respectively) were considered. In the event of absolute humidity loss along the trajectories (for example due to precipitation), the relative contribution to the final humidity was corrected and reduced[79]. For each event, only trajectories which had significant marine moisture uptake with a fraction larger than 0.6 were further assessed (i.e., at least 60% of water in the final humidity is marine source)[53]. The following parameters were reconstructed along the moisture paths based on results from an earlier study showing the strongest correlations with *d-excess* for air moisture[53]: relative humidity at 2 meters with respect to saturation at the sea surface ($RH_{SST}$) and respective temperature at 2 meters minus SST ($T_{2M}$ − SST). Weighted mean $RH_{SST}$ and ($T_{2M}$ − SST) were calculated using $f$ as weights at each time step along a given trajectory and averages of values for all trajectories for a given event were computed. In total, the calculation of backward air parcel trajectories and moisture source parameters was successful for 250 daily rainfall days (out of 263 days where rainfall >15 mm) at Soreq.

To investigate the association of long-term accumulated *d-excess* with moisture uptake conditions and air parcel trajectory location, the backward trajectories were classified according to their position (lon/lat coordinates) and specific humidity content using unsupervised machine learning algorithms. The classification was calculated for 5 days backward trajectories at a 6-hour time resolution as the majority of backward trajectories exhibited moisture uptake in the nearby Mediterranean Sea. The self-organizing maps (SOM) algorithm is an unsupervised neural network used for objective classification and data reduction tasks[82,83], which is achieved by an iterative training process where similar data vectors are grouped into fundamental category maps called the SOM nodes. The finalized SOM nodes represent the classification patterns which best captures the variability of the input data (represented by 25 maps). This classification allows for a topological relationship between the SOM nodes, while similar SOM nodes are found nearby each other, and the largest differences are found at the lattice corners. We grouped similar SOM nodes by applying *k*-means clustering, thus reducing the observations into 4 clusters based on their respective trajectory paths and moisture uptake over the Mediterranean Sea (clusters numbered #0, #1, #2, and #3; Supplementary Fig. 2). 90% of rainfall events (250/277 events) conform to these 4 clusters.

Accumulated parameter (P) cluster averages (for: *d-excess*, $RH_{SST}$, $T_{2M}$ − SST, $\delta^2 H$) were calculated (Eq. 8; based on an adaptation of Eq. 4):

$$P_{cluster} = \frac{\sum P_{event} \times \text{rainfall}}{\sum \text{rainfall}} \qquad (8)$$

Furthermore, we calculated the North Atlantic (NA) cluster frequency ratio (Fig. 6A) as a ratio of the number of rainfall events from both W and NW clusters divided by total number of rainfall events over three-year intervals at Soreq. To investigate rainfall season length corresponding to the clusters we plotted probability histograms binned monthly for the cluster rainfall events (Supplementary Fig. 4).

## Mediterranean Oscillation Index

The $MOI_2$ is defined as the normalized pressure difference between Gibraltar's Northern Frontier (36.1°N, 5.3°W) and Lod Airport in Israel (32.0°N, 34.5°E)[60]. The daily $MOI_2$ was provided by Climatic Research Unit, University of East Anglia (https://crudata.uea.ac.uk/cru/data/moi/) which was calculated using pressure interpolated (16-point Bessel) from NCEP/NCAR reanalysis. The December to February means were calculated for each given rainfall season (Supplementary Data 1) and then averaged over three-year intervals.

## Data availability

All data generated in this study are provided in the Supplementary Information files.

## Code availability

The Supplementary Code for calculating backward trajectory moisture uptake conditions and clusters is provided in the SCfile.zip file.

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

## Acknowledgements

We thank P. Hoppe (MPIC) and A. Sorowke (MPIC) for SEM-EDX, Yvan Rome for fruitful discussions, ECMWF for providing access to ERA-5, and NASA OceanColor for the AQUA-MODIS-terra L3SMI SST data. This research was supported by the Minerva foundation and the Max Planck Society (MPG).

## Author contributions
Conceptualization: E.J.L., M.B.M., A.A., A.M., H.V., A.M.G., V.S. Formal analysis: E.J.L., A.A., M.B.M., V.S., M.S., T.Z., G.Y. Funding acquisition: E.J.L., G.H. Methodology: E.J.L., M.B.M., A.A., H.V., A.M.G., V.S. Resources: G.H., M.B.M., A.A., H.V., A.M.G., S.R.R. Writing—original draft: E.J.L. Writing—review and editing: All authors.

## Funding

## Competing interests
The authors declare no competing interests.
