## [Peer Review File · Nature Communications]

Weakened AMOC related to cooling and atmospheric circulation shifts in the last interglacial Eastern MediterraneanREVIEWER COMMENTS

Reviewer #1 (Remarks to the Author):

The paper is very well written, easy to follow and understand, the goals are clear, the authors present important findings and interesting ideas. However, I have a major problem, namely the paper relies too much on the inclusion-based d-excess data. Why do I have such an opinion? The first warning signal is the unrealistically high d-excess data at about 127 ka. Additionally, the formation temperatures calculated from the $\delta^{18}\text{O}_{\text{fi}}$ and $\delta^{18}\text{O}_{\text{cc}}$ values deviate significantly from the TEX86 temperatures, which was also mentioned by the authors, and as they conclude: „we thus assume that the $\Delta\delta^{18}\text{O}$ temperatures are anomalous.” Either the calcite compositions are diagenetically altered, or the inclusion water $\delta^{18}\text{O}$ values are shifted, or both. However, in speleothems the carbonates are generally free of strong diagenetic influences, and the $\delta^{18}\text{O}_{\text{cc}}$ values are rather difficult to change. Hence, the $\delta^{18}\text{O}_{\text{fi}}$ values become suspicious.

Plotting the $\delta^{18}\text{O}_{\text{fi}}$ and $\delta^2\text{H}$ values, the $\delta^{18}\text{O}_{\text{fi}}$ shifts appear. The samples that yielded anomalously high d-excess values (marked in boxes in the upper panel) are confined to two time periods. These samples show negative $\delta^{18}\text{O}$ shifts from the GMWL, while most of the other samples plot along a usual Mediterranean water line with cca. +20 ‰ d-excess value. The $\delta^{18}\text{O}$ -shifted samples would define a correlation with a slope of 1.3, which is very similar to those found for other speleothems by Demény et al. (2021). Hence I would suspect that the $\delta^{18}\text{O}_{\text{fi}}$ values reflect diagenesis at least in these time periods.

On the other hand, the Peq'in $\delta^2\text{H}$ record is very similar to the Baradla record of Demény et al. (2017), and its pattern is also similar to the TEX86 record's pattern.

Beside using the $\delta^{18}\text{O}_{\text{cc}}$ and $\delta^{18}\text{O}_{\text{fi}}$ values, temperature and d-excess calculations can be conducted in another way. The $\delta^{18}\text{O}_{\text{water}}$ values can be calculated from the $\delta^2\text{H}$ data using the MMWL equation of Goldsmith et al (2017), then the formation temperatures are obtained using the $\delta^{18}\text{O}_{\text{cc}}$ data and a calcite-water oxygen isotope fractionation equation („ $\delta^2\text{H}-\delta^{18}\text{O}_{\text{cc}}$ ”). I used the Kim and O'Neil (1997), the Johnston et al. (2013), and the Daëron et al (2019) equations. The following figure shows the $\delta^2\text{H}-\delta^{18}\text{O}_{\text{cc}}$ temperatures obtained using the Daëron et al (2019) equation, as well as the TEX86 temperatures. The other two equation yielded lower temperatures. The agreement is quite good in my view, indicating that these data can be used for further calculations and discussions.

Finally, I used the TEX86 temperatures, made a linear transformation to 0.1 ka steps (using the PAST software) and gathered interpolated TEX86 temperatures for the inclusion data. Then the $\delta^{18}\text{O}_{\text{water}}$ values were calculated from the $\delta^{18}\text{O}_{\text{cc}}$ values and the Daëron et al (2019) equation, and finally the d-excess values were computed using the $\delta^2\text{H}$ data. The d-excess record is shown below.

The „normal” d-excess range from 15 to 25 ‰, and the rather good agreement with other, independent proxy record patterns would suggest to me that this procedure yields more reliable d-excess data than the direct calculation from the original $\delta^{18}\text{O}_{\text{fi}}$ values.

The interpretations would change a bit, especially for the 129 to 126 ka section, but the 126 to 121 ka section’s interpretation and discussion would not be corrected.

I suggest the authors should consider these ideas and revise the paper, as I think the original $\delta^{18}\text{O}_{\text{fi}}$ values of the older speleothem section (129 to 126 ka) are compromised and can not be used for these calculations. Otherwise the discussion of teleconnection relationships are valuable and interesting for the 126-121 ka section, and if the older section is revised then the paper might be published in Nature Communications (after a second review round).

Some minor comments:

- the petrographic description is good, but some microscopic pictures would be valuable.
- the d-excess values differ from the values that can be obtained using the listed $\delta^2\text{H}$ and $\delta^{18}\text{O}_{\text{fi}}$ values and the usual d-excess equation. This might be related to rounding, but the difference reached 3.6 ‰ for some samples.
- in the Abstract „cave speleothems” is mentioned, but speleothems are cave deposit, so the word „cave” can be omitted.

Reviewer #2 (Remarks to the Author):

Review of "Rapid Mediterranean atmospheric circulation shifts during the last interglacial"

This paper describes the hydroclimate evolution in the Eastern Mediterranean during the last interglacial period (LIG) and discusses potential impacts of the Atlantic Meridional Overturning Circulation (AMOC) on atmospheric patterns over the eastern Mediterranean. The authors reconstruct temperature and surface moisture variability from the isotopic composition of fluid inclusions and GDGTs in stalagmites from Peq'in Cave. The data reveal a stepwise cooling between 126 and 124ka, which is supported by low-resolution analysis (four samples) from Soreq Cave. Coinciding with this cold oscillation, a decrease in d-excess values is interpreted as a change in the moisture source in the Eastern Mediterranean. Higher d-excess values (27.5 ‰) are associated with moisture uptake from the Eastern Mediterranean only and lower values (18.1 ‰) reveals moisture uptake across the whole Mediterranean. Based on this interpretation, the authors suggest a shift in the atmospheric circulation from predominant meridional winds to zonal winds. The authors attribute the change in the regional atmospheric circulation to a slowdown in the AMOC and a decrease in North Atlantic Sea Surface Temperature. To support this hypothesis, the authors compare the episode around 124 ka BP with a 2-year reduction in accumulated rainfall d-excess corresponding with predominant westerlies in the Eastern Mediterranean from 2017 to 2019 that occurred against the backdrop of a weakened AMOC. The primary claim of the manuscript is that a future AMOC weakening trend (which is, by the way, a controversial topic, see Kilbourne et al., 2022_Nature Geoscience, Caesar et al., 2021_Nature Geoscience and Caesar et al., 2022_Nature Geoscience, reply to Kilbourne et al., 2022) may change the rainfall regime in the Eastern Mediterranean.

The manuscript is well-written and well supported by figures, supplementary information and extended data. The methodology applied is novel and adequate for the study. Results and the hypothesis suggested are original, interesting and relevant for the palaeoclimate community. However, I think, there are flaws in the interpretation and conclusion that should be addressed before publication. In general, the study needs to be better supported by additional evidence and the current literature. In particular, a more detailed discussion of the physical mechanisms linking the AMOC slowdown and the change in the wind direction over eastern Mediterranean. So far, the climatic discussion is based on the coincidence of independent events. The manuscript should demonstrate that the interpretation of the results and the comparison of it to other regional records is robust and discuss the shift in atmospheric circulation considering other processes and mechanisms that might influence climate variability in the Eastern Mediterranean. I suggest the following:

1. Improve the regional comparison shown in Figure 2 by adding the published records described in the Supplementary Information 1. Please discuss similar trends but also possible regional differences with, for example, Corchia Cave, one of the best studied speleothem records in the Mediterranean during the LIG, and other marine cores in the Mediterranean and the Iberian Margin. During the climatic oscillation discussed in this manuscript (126-124 ka BP), there are three rapid (centennial to millennial) cold oscillations recorded in Corchia, which are well correlated with North Atlantic cold pulses C27, C27a and C27b. Can you identify these events in your record? I assume, the temporal resolution of the Peq'in Cave does not allow discussion on centennial to millennial timescale? I didn't find information about the sampling resolution of the records presented in this study.
2. There are other palaeoclimate studies that discuss physical mechanisms linking changes in the Greenland ice sheet, the North Atlantic / AMOC, and European and Mediterranean cold events during the LIG, which are not considered in this study. For example, Tzedakis et al (2018_Nature Communications) discuss the role of the subpolar gyre and its impact on mid-latitudes westerlies. Felis et al (2004_Nature) suggest a dominant positive mode of the North Atlantic Oscillation leading to a high-pressure anomaly over the Mediterranean, which favours an anticyclonic flow of surface winds in the eastern Mediterranean resulting in advection of cold air from Europe. Please consider and discuss these mechanisms and show how these processes agree/disagree with the shift to predominant zonal winds that you suggest. Mechanisms behind the increase in temperature and d-excess after 124 ka BP in Peq'in and Soreq Caves should be also discussed.
3. The last section of the manuscript "AMOC related moisture source changes in recent years"

needs to be improved. I like the idea of highlighting the use of the LIG as a possible analogue of future scenarios, but I think the example shown (i.e. changes observed in 2017-2019) is not comparable to the LIG scenario (timescale of the changes, orbital configuration, seasonality, etc). In any case, authors do not discuss any physical mechanisms but report a coincidence of events. As a start point, Wen et al. (2016_Climatic Dynamics) discuss current AMOC-NAO coupling mechanisms in a IPSL-EM5A climate model, which might be useful here.

Other detailed comments are below:

4. Line 17 in the Abstract "this abrupt atmospheric reorganisation was followed by a large cooling that extended beyond Europe into Western Asia". How do the authors demonstrate that the atmospheric reorganisation happened prior to the cooling and it was not the other way around? To confirm this, you would need to synchronise climate records (avoiding wiggle matching) across a regional transect. As a following comment, I have missed information about the chronology of the record, age uncertainty and temporal sampling resolution. Even if this information is published in other papers, a summary of the most relevant information would be appreciated in the supplementary information. The chronological information could be also useful to define the duration of the cold event and abruptness of the transitions.

5. Line 36-38. Please add a section in the Supplementary information describing how the Eastern Mediterranean region is influenced by the AMOC and North Atlantic atmospheric conditions nowadays. I think a conceptual diagram of the currents modes of climate variability influencing the eastern Mediterranean climate on the map shown in Figure 1 would help to visualise the climatic context of the study region.

6. Line 71-83. Please explain what higher / lower d-excess values mean in terms of higher/lower surface moisture and rainfall. It is not clear if the 126-124 ka interval was cold and dry or cold and wet in the Eastern Mediterranean. In figure 2, add an additional y-axis on 2a (temperature) and 2c (surface moisture) showing the reconstructed climate-proxy relationship. E.g. + <--- Temperature - -

7. Methods. In my opinion, the first section of Methods "Pqi'in and Soreq caves and speleothems" should be placed in supplementary information together with chronological information of both records. In the following Methods subheadings, please add information about the sampling resolution.

I hope my comments are useful and help improve a new version of the manuscript.

References:

Caesar et al., 2022. <https://www.nature.com/articles/s41561-022-00897-3>

Caesar et al., 2021. <https://www.nature.com/articles/s41561-021-00699-z>

Kilbourne et al. 2022. <https://doi.org/10.1038/s41561-022-00896-4>

Felis et al., 2004. <https://www.nature.com/articles/nature02546>

Wen et al., 2004. <https://link.springer.com/article/10.1007/s00382-015-2953-y>

(Itemized point-by-point response to all reviewer comments can be found below in *red italic*)

Reply to Reviewer #1

Item #1: The paper is very well written, easy to follow and understand, the goals are clear, the authors present important findings and interesting ideas. However, I have a major problem, namely the paper relies too much on the inclusion-based d-excess data. Why do I have such an opinion? The first warning signal is the unrealistically high d-excess data at about 127 ka.

We thank the reviewer for the positive response following reading our manuscript. Regarding the high d-excess values at ca. 127 ka, however, we feel we must disagree, as d-excess values of up to ~35‰ are certainly not abnormal. This is, for example, shown by the range of d-excess values for the Last Interglacial from fluid inclusion stable isotopes at Soreq Cave by Matthews et al., (2021). In that paper, not only the measured fluid inclusion d-excess values, but also the d-excess values calculated by combining clumped isotope derived $\delta^{18}\text{O}_{\text{water}}$ and fluid inclusion $\delta^2\text{H}$ data show elevated values (see page 9 lines 3 to 9 in revised manuscript). The LIG d-excess range is also acceptable with respect to today as evident in moisture above the Mediterranean Sea (Gat et al., 2003), air vapor in Israel (Angert et al., 2008) and rainfall at Peqi'in (Ayalon et al., 2004). Below are box-whisker charts of selected data and a table after Ayalon et al. (2004) to illustrate that fluid inclusion d-excess (nor associated low $\delta^{18}\text{O}$) are anomalous:

Date	mm	d18O	dD	d-excess
05/12/01	13.2	-10.61	-58.8	26.08
07/01/02	178	-9.18	-44.3	29.14
10/01/02	34.2	-9.85	-50.2	28.6
21/12/02	12.52	-10.24	-52.3	29.62
25/12/02	28.3	-7.92	-36.4	26.96
29/01/03	175.6	-8.71	-35.2	34.48
04/02/03	87.5	-8.36	-36.8	30.08
05/02/03	50.6	-7.92	-36.7	26.66
23/02/03	40	-8.58	-32.7	35.94
26/02/03	22	-8.62	-32.4	36.56
20/03/03	124.1	-8.3	-31.3	35.1

The selected Peqi'in rainfall data from Ayalon et al. (2004) overlap with early-LIG fluid inclusion water. They were taken during only two rainfall seasons (2001 -2003; see table below), however, there are many more instances like this for other years and other rainfall monitoring sites.

Item #2: Additionally, the formation temperatures calculated from the $\delta^{18}\text{O}_{\text{fi}}$ and $\delta^{18}\text{O}_{\text{cc}}$ values deviate significantly from the TEX₈₆ temperatures, which was also mentioned by the authors, and as they conclude: „we thus assume that the $\delta^{18}\text{O}$ temperatures are anomalous.” Either the calcite compositions are diagenetically altered, or the inclusion water $\delta^{18}\text{O}$ values are shifted, or both. However, in speleothems the carbonates are generally free of strong diagenetic influences, and the $\delta^{18}\text{O}_{\text{cc}}$ values are rather difficult to change. Hence, the $\delta^{18}\text{O}_{\text{fi}}$ values become suspicious.

Ideally the fluid inclusion isotope system works so that reliable temperatures result from isotope equilibrium (or near equilibrium) calculation of paired calcite and FI $\delta^{18}\text{O}$. However, in reality significant deviation from expected equilibrium temperatures occur (Daëron et al., 2019). In the data, as we present them, the temperatures calculated from the fluid inclusions are generally lower than or equal to the TEX₈₆ temperatures during the LIG (Supplementary Fig. 1). Only during the early-LIG do the TEX₈₆ and $\delta^{18}\text{O}$ calcite-FI temperatures significantly deviate. Diagenetic effects (recrystallization) on fluid inclusion $\delta^{18}\text{O}$ are indeed a factor of concern. However, lower than expected $\delta^{18}\text{O}$ calcite-FI temperatures may also result from precipitation of calcite under disequilibrium (e.g. following prior calcite precipitation; Deininger et al., 2021) or seasonal bias in fluid inclusion capture (e.g. Wassenburg et al 2021) (see Supplementary Information Text).

Our reasoning that d-excess values of ~35‰ are expected, rather than anomalous, in addition to the good correlation of an early-LIG $\delta^{18}\text{O}$ minima to other regional speleothem and marine records (Fig. 3), weighs heavily in our preference for the scenario that the lower-than-expected temperatures are caused either by disequilibrium calcite precipitation or seasonality imprinted on the speleothem fabric, rather than diagenetic alteration of the fluid inclusion water (see page 7 line 3 to page 8 line 15 in

revised manuscript with tracked changes). We therefore explain the fluid inclusion d-excess record as it is but also consider, as you suggest, a potential shift of FI $\delta^{18}\text{O}$ due to alteration (Demény et al., 2016). In the revised manuscript we include an alternate calculated d-excess record by incorporating the fluid inclusion water $\delta^2\text{H}$, calcite $\delta^{18}\text{O}$, TEX_{86} temperatures, and the equation provided by Daëron et al., (2019) (see eq. 4; $d\text{-excess}_{\text{calc}}$. Fig. 3F; see our reply to Item #5 below).

Item #3:

Plotting the $\delta^{18}\text{O}_{\text{fi}}$ and $\delta^2\text{H}$ values, the $\delta^{18}\text{O}_{\text{fi}}$ shifts appear. The samples that yielded anomalously high d-excess values (marked in boxes in the upper panel) are confined to two time periods. These samples show negative $\delta^{18}\text{O}$ shifts from the GMWL, while most of the other samples plot along a usual Mediterranean water line with cca. +20 ‰ d-excess value. The $\delta^{18}\text{O}$ -shifted samples would define a correlation with a slope of 1.3, which is very similar to those found for other speleothems by Demény et al. (2021). Hence I would suspect that the $\delta^{18}\text{O}_{\text{fi}}$ values reflect diagenesis at least in these time periods.

Thank you for taking the time to replot the data and for providing us with these suggestions. Please see our reply to item #1 and #2.

Item #4:

On the other hand, the Peq'in $\delta^2\text{H}$ record is very similar to the Baradla record of Demény et al. (2017), and its pattern is also similar to the TEX_{86} record's pattern.

Yes indeed, and this would strengthen the paleo-climate significance of these fluid inclusion records (see page 8 lines 16 to 21 in revised manuscript).

Item #5:

Beside using the $\delta^{18}\text{O}_{\text{cc}}$ and $\delta^{18}\text{O}_{\text{fi}}$ values, temperature and d-excess calculations can be conducted in another way. The $\delta^{18}\text{O}_{\text{water}}$ values can be calculated from the $\delta^2\text{H}$ data using the MMWL equation of Goldsmith et al (2017), then the formation temperatures are obtained using the $\delta^{18}\text{O}_{\text{cc}}$ data and a calcite-water oxygen isotope fractionation equation („ $\delta^2\text{H}-\delta^{18}\text{O}_{\text{cc}}$ ”). I used the Kim and O’Neil (1997), the Johnston et al. (2013), and the Daëron et al (2019) equations. The following figure shows the $\delta^2\text{H}-\delta^{18}\text{O}_{\text{cc}}$ temperatures obtained using the Daëron et al (2019) equation, as well as the TEX86 temperatures. The other two equation yielded lower temperatures. The agreement is quite good in my view, indicating that these data can be used for further calculations and discussions.

Finally, I used the TEX86 temperatures, made a linear transformation to 0.1 ka steps (using the PAST software) and gathered interpolated TEX86 temperatures for the inclusion data. Then the $\delta^{18}\text{O}_{\text{water}}$ values were calculated from the $\delta^{18}\text{O}_{\text{cc}}$ values and the Daëron et al (2019) equation, and finally the d-excess values were computed using the $\delta^2\text{H}$ data. The d-excess record is shown below.

The „normal” d-excess range from 15 to 25 ‰, and the rather good agreement with other, independent proxy record patterns would suggest to me that this procedure yields more reliable d-excess data than the direct calculation from the original $\delta^{18}\text{O}_{\text{fi}}$ values. The interpretations would change a bit, especially for the 129 to 126 ka section, but the 126 to 121 ka section’s interpretation and discussion would not be corrected.

We thank you for these important suggestions. Matthews et al. (2021) demonstrated, that the MWL in the Eastern Mediterranean can change significantly. Thus, we cannot assume the MMWL equation of Goldsmith et al. (2017) for the LIG and we did not calculate the paleo-temperatures as suggested. However, we did calculate a d-excess record using calcite $\delta^{18}\text{O}$, fluid inclusion water $\delta^2\text{H}$, the TEX_{86} temperatures and the Daëron et al (2019) equation (eq. 4; Fig. 3F). The $d\text{-excess}_{\text{calc.}}$ record shows a flatter record during the early-LIG which is a direct reflection of the early-LIG high calcite $\delta^{18}\text{O}$ at Peqi’in. Nevertheless the $d\text{-excess}_{\text{calc.}}$ record reveals a minimum at ca. 124 ka, as does the ‘original’ fluid inclusion d-excess record. Thus, our suggestion that the mid-LIG cooling was characterized by more zonal winds over the Mediterranean and associated rainfall does not change.

Item #5:

I suggest the authors should consider these ideas and revise the paper, as I think the original $\delta^{18}\text{O}_{\text{fi}}$ values of the older speleothem section (129 to 126 ka) are compromised and can not be used for these calculations. Otherwise the discussion of teleconnection relationships are valuable and interesting for the 126-121 ka section, and if the older section is revised then the paper might be published in Nature Communications (after a second review round).

While we understand your reasoning, after carefully weighing all the available data and theoretical considerations we remain of the opinion that diagenetic alteration is not needed to explain the pattern of measured d-excess values nor fluid inclusion $\delta^{18}\text{O}$ (See our reply to items #1 and #2; Supplementary Information Text).

Still, we understand that we cannot be 100% sure of our interpretation, as we fully agree that fluid inclusion water is in principle sensitive to alteration. We believe that the addition of the new $d\text{-excess}_{\text{calc.}}$ as the reviewer suggests, allows the reader to appreciate the uncertainties around our interpretation of the mid-LIG teleconnections with the weakened-AMOC. We are therefore very grateful for his/her detailed comments and suggestions in this matter.

Item #6:

- the petrographic description is good, but some microscopic pictures would be valuable.

Agreed. In the revised manuscript we have added thin section images (see Supplementary Information Fig. 6).

Item #7:

- the d-excess values differ from the values that can be obtained using the listed $\delta^2\text{H}$ and $\delta^{18}\text{O}_{\text{fi}}$ values and the usual d-excess equation. This might be related to rounding, but the difference reached 3.6 ‰ for some samples.

Thanks for noticing. We have reviewed the data files and corrected for all inconsistencies.

Item #8:

- in the Abstract „cave speleothems” is mentioned, but speleothems are cave deposits, so the word „cave” can be omitted.

Agreed and omitted.

Reply to Reviewer #2

Item #9: In general, the study needs to be better supported by additional evidence and the current literature. In particular, a more detailed discussion of the physical mechanisms linking the AMOC slowdown and the change in the wind direction over eastern Mediterranean. So far, the climatic discussion is based on the coincidence of independent events.

We thank the reviewer for the positive response to our manuscript. In the revised manuscript we include a discussion of the physical mechanisms linking weakening of the AMOC and changes in the westerlies and wind direction in the Eastern Mediterranean during the LIG. Specifically, we refer to the LIG freshwater discharge experiments in Tzedakis et al., (2018) (see page 13 line 12 to page 14 line 2 in the revised manuscript with tracked changes). We hope that with higher resolution models in the future we can gain further insight into the atmospheric changes local to the Eastern Mediterranean.

Regarding modern: in the revised manuscript we now acknowledge the debate regarding the recent AMOC weakening, and include your suggested references (see page 3 line 2 to 4 in revised manuscript). We also include a more substantial review of rainfall regime for this region (see page 10 line 2 to 13 in revised manuscript). Rather than describing a direct teleconnection mechanism between Eastern Mediterranean rainfall and a weakening AMOC, alongside the modern decreasing d-excess trend we present an DJF 3yr average of the Mediterranean Oscillation Index which provides support for a change in the position of Cyprus Low cyclones in the Eastern Mediterranean (Ziv et al. 2014; Drori et al. 2021; see Fig. 6B; page 14 line 21 to page 15 line 7). With respect to the 'possible' AMOC-Eastern Mediterranean climate teleconnection: we have reduced speculations and now merely raise it is a question (see page 14 line 3; page 15 line 9 to 17).

Item #10: The manuscript should demonstrate that the interpretation of the results and the comparison of it to other regional records is robust and discuss the shift in atmospheric circulation considering other processes and mechanisms that might influence climate variability in the Eastern Mediterranean. I suggest the following:

1. Improve the regional comparison shown in Figure 2 by adding the published records described in the Supplementary Information 1.

Agreed. We now include this supplementary figure and the accompanying description of these records in the revised main manuscript (see Fig. 2; page 5 line 10 to page 6 line 15).

Item 11: Please discuss similar trends but also possible regional differences with, for example, Corchia Cave, one of the best studied speleothem records in the

Mediterranean during the LIG, and other marine cores in the Mediterranean and the Iberian Margin.

In the revised manuscript we now present the Corchia Cave $\delta^{18}\text{O}$ record (see Fig. 3D) and, along with the temperate pollen record from core MD01-2444 (Portuguese margin), we discuss this in the context of the Eastern Mediterranean (see page 8 lines 9 to 15 in revised manuscript). We also show the LIG $\delta^{18}\text{O}$ from G. Ruber taken from core LC21 in the Eastern Mediterranean as it is highly relevant (see Fig. 3C; and page 7 lines 5 to 11 in revised manuscript).

Item 12: During the climatic oscillation discussed in this manuscript (126-124 ka BP), there are three rapid (centennial to millennial) cold oscillations recorded in Corchia, which are well correlated with North Atlantic cold pulses C27, C27a and C27b. Can you identify these events in your record? I assume, the temporal resolution of the Peq'in Cave does not allow discussion on centennial to millennial timescale? I didn't find information about the sampling resolution of the records presented in this study.

The temporal resolution here is limited to identify these pulses. We include a description of sampling resolution in the revised manuscript and supplementary data (Sample error bars in Fig. 2 and 3; see page 17 lines 3 to 6 in the revised manuscript).

Item #13: 2. There are other palaeoclimate studies that discuss physical mechanisms linking changes in the Greenland ice sheet, the North Atlantic / AMOC, and European and Mediterranean cold events during the LIG, which are not considered in this study. For example, Tzedakis et al (2018_ Nature Communications) discuss the role of the subpolar gyre and its impact on mid-latitudes westerlies. Felis et al (2004_ Nature) suggest a dominant positive mode of the North Atlantic Oscillation leading to a high-pressure anomaly over the Mediterranean, which favours an anticyclonic flow of surface winds in the eastern Mediterranean resulting in advection of cold air from Europe. Please consider and discuss these mechanisms and show how these processes agree/disagree with the shift to predominant zonal winds that you suggest.

Thank you for these suggestions. In the revised manuscript we discuss the Eastern Mediterranean climate associated with a weakened AMOC during the LIG from the North Atlantic freshwater discharge experiments by Tzedakis et al., (2018) (see page 3 lines 13 to 21; page 13 line 12 to page 14 line 2; see also our reply to item #9). Regarding the North Atlantic Oscillation: while Felis et al., (2004) argues that the climate in the Northern Red Sea is strongly affected by the NAO, there are several studies suggesting weak modern correlation between the NAO and rainfall (Ziv et al. 2014). We thus refrain from discussing the positive-NAO mode in the LIG suggested by Felis et al., (2004).

For the modern, we present the Mediterranean Oscillation Index (MOI_2) which is calculated as the pressure difference between Gibraltar and Lod (Israel). The MOI_2 has been shown to correlate well with the position of wintertime storm tracks in Israel (Fig. 6B).

Item #14: Mechanisms behind the increase in temperature and δ -excess after 124 ka BP in Piq'in and Soreq Caves should be also discussed.

In the revised manuscript we discuss the trends over the entire LIG for the TEX_{86} (page 6 lines 6 to 11) and FI data (page 8 lines 16 to 18; page 9 lines 13 to 14) and mechanisms (see page 13 line 23 to page 14 line 2; see also our reply to item #9).

Item #15: 3. The last section of the manuscript “AMOC related moisture source changes in recent years” needs to be improved. I like the idea of highlighting the use of the LIG as a possible analogue of future scenarios, but I think the example shown (i.e. changes observed in 2017-2019) is not comparable to the LIG scenario (timescale of the changes, orbital configuration, seasonality, etc). In any case, authors do not discuss any physical mechanisms but report a coincidence of events. As a start point, Wen et al. (2016_Climatic Dynamics) discuss current AMOC-NAO coupling mechanisms in a IPSL-EM5A climate model, which might be useful here.

We have rewritten large parts the last section (see page 14 line 4 to 9; page 14 line 21 to page 15 line 17 in the revised manuscript). As we mentioned in our reply to items #9 and #13, we now include the MOI_2 (Fig. 7B) which supports a shift in the position of Eastern Mediterranean cyclones and change in moisture uptake location as suggested by the δ -excess (see also reply to Item #9 and, with respect to the NAO, Item #13 above).

Item #16: 4. Line 17 in the Abstract “this abrupt atmospheric reorganisation was followed by a large cooling that extended beyond Europe into Western Asia”. How do the authors demonstrate that the atmospheric reorganisation happened prior to the cooling and it was not the other way around? To confirm this, you would need to synchronise climate records (avoiding wiggle matching) across a regional transect.

Agreed and removed.

Item #17: I have missed information about the chronology of the record, age uncertainty and temporal sampling resolution. Even if this information is published in other papers, a summary of the most relevant information would be appreciated in the supplementary information. The chronological information could be also useful to define the duration of the cold event and abruptness of the transitions.

In the revised manuscript we include the chronology of the record (Supplementary Table 1; Figure 2 at the bottom), age uncertainties, and the temporal sampling resolution (page 16 line 17 to page 17 line 6; Supplementary Data).

Item #18: 5. Line 36-38. Please add a section in the Supplementary information describing how the Eastern Mediterranean region is influenced by the AMOC and North Atlantic atmospheric conditions nowadays. I think a conceptual diagram of the currents

modes of climate variability influencing the eastern Mediterranean climate on the map shown in Figure 1 would help to visualise the climatic context of the study region.

In the revised manuscript we include a broader overview of the Eastern Mediterranean rainfall regime (see page 10 line 2 to 13 in revised manuscript; Fig. 1B) and a description of teleconnection patterns with the MOI₂ (see page 14 line 21 to page 15 line 7 in revised manuscript). Further investigation into the influence of the AMOC on the Eastern Mediterranean rainfall regime is required (see page 15 line 13 to 17)

Item #19: 6. Line 71-83. Please explain what higher / lower d-excess values mean in terms of higher/lower surface moisture and rainfall. It is not clear if the 126-124 ka interval was cold and dry or cold and wet in the Eastern Mediterranean. In figure 2, add an additional y-axis on 2a (temperature) and 2c (surface moisture) showing the reconstructed climate-proxy relationship. E.g. + <--- Temperature - -

While the d-excess is a faithful measure of the humidity gradient and location of moisture uptake this measurement does not reflect the amount of rainfall in the region which is more complex.

Item #20: 7. Methods. In my opinion, the first section of Methods “Pqi’in and Soreq caves and speleothems” should be placed in supplementary information together with chronological information of both records. In the following Methods subheadings, please add information about the sampling resolution.

All the methods are in the methods section according to Nat. Commun. request. We have added the chronology as a supplementary table and provide information on sampling resolution to the methods (see our reply to item #17, above).

References:

*Angert, A., Lee, J.E. & Yakir, D. Seasonal variations in the isotopic composition of near-surface water vapour in the eastern Mediterranean. Tellus B: Chemical and Physical Meteorology, **60**, 674-684. (2008)*

Ayalon, A., Bar-Matthews, M. & Schilman, B. Rainfall isotopic characteristics at various sites in Israel and the relationships with unsaturated zone water (GSI/16/04, Geological Survey of Israel, 2004).

*Daëron, M., Drysdale, R.N., Peral, M., Huyghe, D., Blamart, D., Coplen, T.B., Lartaud, F. & Zanchetta, G. Most Earth-surface calcites precipitate out of isotopic equilibrium. Nat. Commun., **10**, p.429 (2019).*

*Deininger, M. et al. Are oxygen isotope fractionation factors between calcite and water derived from speleothems systematically biased due to prior calcite precipitation (PCP)? Geochim. Cosmochim. Ac., **305**, 212-227 (2021).*

Demény, A. Recrystallization-induced oxygen isotope changes in inclusion-hosted water of speleothems – paleoclimatological implications. *Quatern. Int.* **415**, 25-32 (2016).

Drori, R., Ziv, B., Saaroni, H., Etkin, A. & Sheffer, E. Recent changes in the rain regime over the Mediterranean climate region of Israel. *Climatic Change*, **167**, 1-2 (2021).

Felis, T., Lohmann, G., Kuhnert, H., Lorenz, S.J., Scholz, D., Pätzold, J., Al-Rousan, S.A. & Al-Moghrabi, S.M. Increased seasonality in Middle East temperatures during the last interglacial period. *Nature* **429**, 164-168 (2004).

Gat, J. R. & Carmi, I. Effect of climate changes on the precipitation patterns and isotopic composition of water in a climate transition zone: case of the eastern Mediterranean Sea area. *IAHS Spec. Publ.* **168**, 513-523 (1987).

Goldsmith, Y. et al. The modern and Last Glacial Maximum hydrological cycles of the Eastern Mediterranean and the Levant from a water isotope perspective. *Earth Plan. Sci. Lett.* **457**, 302-312 (2017).

Matthews, A., Affek, H. P., Ayalon, A., Vonhof, H. B. & Bar-Matthews, M. Eastern Mediterranean climate change deduced from the Soreq Cave fluid inclusion stable isotopes and carbonate clumped isotopes record of the last 160 ka. *Quat. Sci. Rev.* **272**, 107223 (2021).

Tzedakis, P. C. et al. Enhanced climate instability in the North Atlantic and southern Europe during the Last Interglacial. *Nat. Commun.* **9**, 1–4 (2018).

Wassenburg, J. A. et al. Penultimate deglaciation Asian monsoon response to North Atlantic circulation collapse. *Nat. Geosci.* **14**, 937-941 (2021).

Ziv, B., Saaroni, H., Pargament, R., Harpaz, T., & Alpert, P. Trends in rainfall regime over Israel, 1975-2010, and their relationship to large-scale variability. *Reg. Environ. Chang.* **14**, 1751–1764 (2014)

REVIEWERS' COMMENTS

Reviewer #1 (Remarks to the Author):

I went through the revised manuscript. Although our interpretations of diagenetic influences and signs may differ, the revised manuscript addressed this problem. The changes are appropriate and the paper has been improved.

Reviewer #2 (Remarks to the Author):

The revisions have addressed my major concerns and the manuscript has improved considerably. I believe this paper will be an important contribution to the palaeoclimate community. Congratulations to the authors.

As a last comment, I think the abstract and the introduction may provide a wrong impression/idea of what the paper is about. Reading these two sections, it seems like the paper is going to provide insights into the mechanisms linking AMOC and atmospheric dynamics in the North Atlantic region. However, the study discusses potential impacts of AMOC on Mediterranean climate dynamics and how a AMOC slowdown makes the Eastern Mediterranean more sensitive to mid-latitude westerlies. I suggest, minor changes in the wording only. For example:

Line 5 in the abstract: changes that accompany an AMOC slowdown BEYOND THE NORTH ATLANTIC REALM.

Lin 16 in the abstract: It needs an additional sentence explaining the link between the shift to zonal winds and the AMOC slowdown,.

Line 21 in the introduction: I prefer the previous version where you used "teleconnection patterns" in this sentence. I suggest: while LIG climate models suggest hemispheric climate changes associated with a weakened AMOC, so far, there HAVE been not robust reconstructions SHOWING EVIDENCE FOR TELECONNECTION PATTERNS LINKING AMOC SLOWDOWN WITH MEDITERRANEAN COOLING.

There are also a few typos through the text that need to be checked. There are special characters and may be an issue with the pdf version.

Best,

Celia Martin-Puertas

Our response to reviewer #2 comments (***bold italic font***)

1.

“As a last comment, I think the abstract and the introduction may provide a wrong impression/idea of what the paper is about. Reading these two sections, it seems like the paper is going to provide insights into the mechanisms linking AMOC and atmospheric dynamics in the North Atlantic region. However, the study discusses potential impacts of AMOC on Mediterranean climate dynamics and how a AMOC slowdown makes the Eastern Mediterranean more sensitive to mid-latitude westerlies. I suggest, minor changes in the wording only. For example:

Line 5 in the abstract: changes that accompany an AMOC slowdown BEYOND THE NORTH ATLANTIC REALM.”

Reply: We agree and have added your suggestion to the abstract (see line 4, page 2 in revised manuscript).

2.

“Lin 16 in the abstract: It needs an additional sentence explaining the link between the shift to zonal winds and the AMOC slowdown,.”

Reply: We were requested to cut 72 excess words from the abstract and thus, unfortunately, we could not add a new sentence. Instead, we connect the previous sentence to the AMOC (see line 11, page 2 in revised manuscript).

3.

Line 21 in the introduction: I prefer the previous version where you used "teleconnection patterns" in this sentence. I suggest: while LIG climate models suggest hemispheric climate changes associated with a weakened AMOC, so far, there HAVE been not robust reconstructions SHOWING EVIDENCE FOR TELECONNECTION PATTERNS LINKING AMOC SLOWDOWN WITH MEDITERRANEAN COOLING.”

Reply: We included this sentence in the introduction (see line 22-24, page 3 in revised manuscript).

4.

There are also a few typos through the text that need to be checked. There are special characters and may be an issue with the pdf version.

Reply: Thank you. We checked the text for typos and the format of the special characters.